# Beyond Regrets:
# Geometric Metrics for Bayesian Optimization

## Abstract

Bayesian optimization is a principled optimization strategy for a black-box objective function. It shows its effectiveness in a wide variety of real-world applications such as scientific discovery and experimental design. In general, the performance of Bayesian optimization is reported through regret-based metrics such as instantaneous, simple, and cumulative regrets. These metrics only rely on function evaluations, so that they do not consider geometric relationships between query points and global solutions, or query points themselves. Notably, they cannot discriminate if multiple global solutions are successfully found. Moreover, they do not evaluate Bayesian optimization's abilities to exploit and explore a search space given. To tackle these issues, we propose four new geometric metrics, i.e., precision, recall, average degree, and average distance. These metrics allow us to compare Bayesian optimization algorithms considering the geometry of both query points and global optima, or query points. However, they are accompanied by an extra parameter, which needs to be carefully determined. We therefore devise the parameter-free forms of the respective metrics by integrating out the additional parameter. Finally, we empirically validate that our proposed metrics can provide more delicate interpretation of Bayesian optimization algorithms, on top of assessment via the conventional metrics.

## 1 Introduction

Bayesian optimization (Garnett, 2023) is a principled optimization strategy with a probabilistic regression model for a black-box objective function. It demonstrates its effectiveness and validity in a wide variety of real-world applications such as scientific discovery (Griffiths & Hernández-Lobato, 2020; Kim et al., 2022; 2024) and experimental design (Attia et al., 2020; Shields et al., 2021; Ament et al., 2023). These applications sometimes require not only finding a global optimum in terms of function evaluations but also diversifying the candidates of global solutions on a search space given. Notably, such a need to find diversified solution candidates is demanded in many real-world scientific problems such as virtual screening (Graff et al., 2021; Fromer et al., 2024) and materials discovery (Tamura et al., 2021; Maus et al., 2023).

To measure the performance of Bayesian optimization, we generally utilize instantaneous, simple, and cumulative regrets; each definition can be found in Section 2. These performance metrics all focus on measuring how the function value of some query point is close to the function value of a global solution (or potentially global solutions). However, they only assess a solution quality in terms of function evaluations, so that they do not consider relationships between query points and global solutions, or query points themselves. In particular, if the finding of multiple global optima is crucial, these regret-based metrics often fail in discriminating global solution candidates that can diversify the candidates. In addition, it is difficult to evaluate the exploitation and exploration abilities of Bayesian optimization.

In this paper, we propose new geometric metrics beyond the regret-based metrics for Bayesian optimization, in order to mitigate the drawbacks of the conventional metrics and improve the understanding of Bayesian optimization algorithms. In particular, four categories of our new metrics, i.e., *precision*, *recall*, *average degree*, and *average distance*, are designed to evaluate how many query points are close to multiple global solutions, and vice versa, or how query points are distributed. However, these metrics are accompanied by an

Table 1: Comparisons between metrics for Bayesian optimization supposing that the corresponding metric is computed at iteration $t$ where $s$ is the number of global optima. GO Info., Multiple GO, and Additional Parameter indicate a requirement of global optimum information, consideration to multiple global optima, and need for an additional parameter, respectively. $\mathbf{x}$ and $\mathbf{x}^*$ represent a query point and a global optimum, respectively, and PF stands for parameter-free.

| Metric | Inputs | GO Info. | Multiple GO | Additional Parameter |
|---|---|---|---|---|
| Instantaneous regret | $f(\mathbf{x}_t), f(\mathbf{x}^*)$ | ✓ | | |
| Simple regret | $f(\mathbf{x}_1), f(\mathbf{x}_2), \ldots, f(\mathbf{x}_t), f(\mathbf{x}^*)$ | ✓ | | |
| Cumulative regret | $f(\mathbf{x}_1), f(\mathbf{x}_2), \ldots, f(\mathbf{x}_t), f(\mathbf{x}^*)$ | ✓ | | |
| Precision | $\mathbf{x}_1, \mathbf{x}_2, \ldots, \mathbf{x}_t, \mathbf{x}_1^*, \mathbf{x}_2^*, \ldots, \mathbf{x}_s^*$ | ✓ | ✓ | ✓ |
| Recall | $\mathbf{x}_1, \mathbf{x}_2, \ldots, \mathbf{x}_t, \mathbf{x}_1^*, \mathbf{x}_2^*, \ldots, \mathbf{x}_s^*$ | ✓ | ✓ | ✓ |
| Average degree | $\mathbf{x}_1, \mathbf{x}_2, \ldots, \mathbf{x}_t$ | | | ✓ |
| Average distance | $\mathbf{x}_1, \mathbf{x}_2, \ldots, \mathbf{x}_t$ | | | ✓ |
| PF precision | $\mathbf{x}_1, \mathbf{x}_2, \ldots, \mathbf{x}_t, \mathbf{x}_1^*, \mathbf{x}_2^*, \ldots, \mathbf{x}_s^*$ | ✓ | ✓ | |
| PF recall | $\mathbf{x}_1, \mathbf{x}_2, \ldots, \mathbf{x}_t, \mathbf{x}_1^*, \mathbf{x}_2^*, \ldots, \mathbf{x}_s^*$ | ✓ | ✓ | |
| PF average degree | $\mathbf{x}_1, \mathbf{x}_2, \ldots, \mathbf{x}_t$ | | | |
| PF average distance | $\mathbf{x}_1, \mathbf{x}_2, \ldots, \mathbf{x}_t$ | | | |

additional parameter. For example, precision, recall, and average degree are needed to determine the size of ball query, and average distance requires the selection of the number of nearest neighbors we are interested in. To alleviate the need to choose the additional parameter, the parameter-free forms of the respective metrics are also proposed. The additional parameter is integrated out by sampling it from a particular distribution. The characteristics of diverse metrics are compared in Table 1.

To explain the procedure of Bayesian optimization briefly, we cover Bayesian optimization first.

**Bayesian Optimization.** We are supposed to solve the following minimization problem:

$$\mathbf{x}^+ = \arg\min_{\mathbf{x} \in \mathcal{X}} f(\mathbf{x}), \tag{1}$$

where $\mathcal{X} \subset \mathbb{R}^d$ is a $d$-dimensional search space and $f$ is a black-box function. We deal with $f$ using a history of query points and their evaluations in Bayesian optimization, so that we rely on a probabilistic regression model to estimate the unknown function. In particular, we can choose one of generic probabilistic regression models such as Gaussian processes (Rasmussen & Williams, 2006), Student-$t$ processes (Shah et al., 2014; Martinez-Cantin et al., 2018), deep neural networks (Snoek et al., 2015), Bayesian neural networks (Springenberg et al., 2016; Li et al., 2023), and tree-based regression models (Hutter et al., 2011; Kim & Choi, 2022). These regression models are capable of producing function value estimates over query points in a search space and their uncertainty estimates, which are able to be used to balance exploitation and exploration. By employing the regression models, an acquisition function to sample the next solution candidate is defined. Probability of improvement (Kushner, 1964), expected improvement (Jones et al., 1998), and Gaussian process upper confidence bound (Srinivas et al., 2010) are often used as an acquisition function. Bayesian optimization eventually determines the next query point by maximizing the acquisition function. We can sequentially find a better solution repeating these steps in determining and evaluating the query point. By the sample-efficient nature of Bayesian optimization, it is beneficial for optimizing a black-box function that is costly to evaluate.

Before describing the details of metrics for Bayesian optimization, we summarize the contributions of this work as follows:

(i) We identify the drawbacks of the conventional regret-based metrics, i.e., instantaneous, simple, and cumulative regrets, for Bayesian optimization;

(ii) We propose four geometric metrics, i.e., precision, recall, average degree, and average distance, and their parameter-free forms;

(iii) We show that our new metrics help us to interpret and understand the performance of Bayesian optimization from distinct perspectives.

*We will release our implementation upon publication.*

## 2 Instantaneous, Simple, and Cumulative Regrets

In Bayesian optimization, instantaneous, simple, and cumulative regrets are often used to report its performance (Garnett, 2023). Intriguingly, these metrics are suitable for not only evaluating the performance of optimization algorithms but also conducting their theoretical analysis. Given a global optimum $\mathbf{x}^*$ and a query point $\mathbf{x}_t$ at iteration $t$, an instantaneous regret is defined as the following:

$$r_t^{\text{ins}} = f(\mathbf{x}_t) - f(\mathbf{x}^*). \tag{2}$$

It is noteworthy that this metric does not need to take into account multiple global optima because function evaluations are only involved in the metric. The instantaneous regret (2) represents the quality of every immediate decision. Due to the property of a trade-off between exploitation and exploration in Bayesian optimization, a series of instantaneous regrets in a single round tends to oscillate. Such oscillation comes from exploration-focused and exploitation-focused decisions; rigorously speaking, it is hard to identify whether a particular function evaluation is led by exploration-focused or exploitation-focused decision, but a dominant factor, i.e., either exploration or exploitation, may exist when a query point is selected. Moreover, while the oscillation of instantaneous regrets does not imply that exploration-focused evaluations always differ from exploitation-focused ones, exploration-focused decisions are likely to obtain unexpected function evaluations that are far from the best evaluation until the current iteration. For these reasons, it is unlikely to consider the instantaneous regret as a solid performance metric to show the performance of Bayesian optimization. As a straightforward alternative to the instantaneous regret, a simple regret is defined as the following:

$$r_t^{\text{sim}} = \min_{i=1}^{t} f(\mathbf{x}_i) - f(\mathbf{x}^*), \tag{3}$$

where $\mathbf{x}_1, \ldots, \mathbf{x}_t$ are $t$ query points. The simple regret (3) is equivalent to the minimum of instantaneous regrets until $t$ iterations, which can represent the best solution out of $t$ solution candidates $\mathbf{x}_1, \mathbf{x}_2, \ldots, \mathbf{x}_t$. Unlike the instantaneous regret, the simple regret is monotonically decreasing by the definition of (3). The common goal of Bayesian optimization is to seek a solution that makes (3) zero, which implies that the best solution found by Bayesian optimization coincides with a global solution; if there exist multiple global solutions, we cannot verify that all global solutions are sought by Bayesian optimization with this metric. Furthermore, using (2), we can readily define a cumulative regret until iteration $t$:

$$R_t = \sum_{i=1}^{t} r_i^{\text{ins}}, \tag{4}$$

which is also used as the metric of Bayesian optimization. Since $r_i^{\text{ins}} \geq 0$, the cumulative regret is always monotonically increasing.

Although instantaneous, simple, and cumulative regrets are representative metrics for empirically showing the performance of Bayesian optimization and theoretically analyzing Bayesian optimization algorithms (Garnett, 2023), it is challenging to consider geometric relationships between query points and global solutions, or query points themselves, verify whether multiple global optima are successfully discovered or not, and measure the degrees of exploration and exploitation. To tackle the aforementioned challenges and provide the better understanding of Bayesian optimization outcomes, we propose new geometric metrics for Bayesian optimization in Section 3.

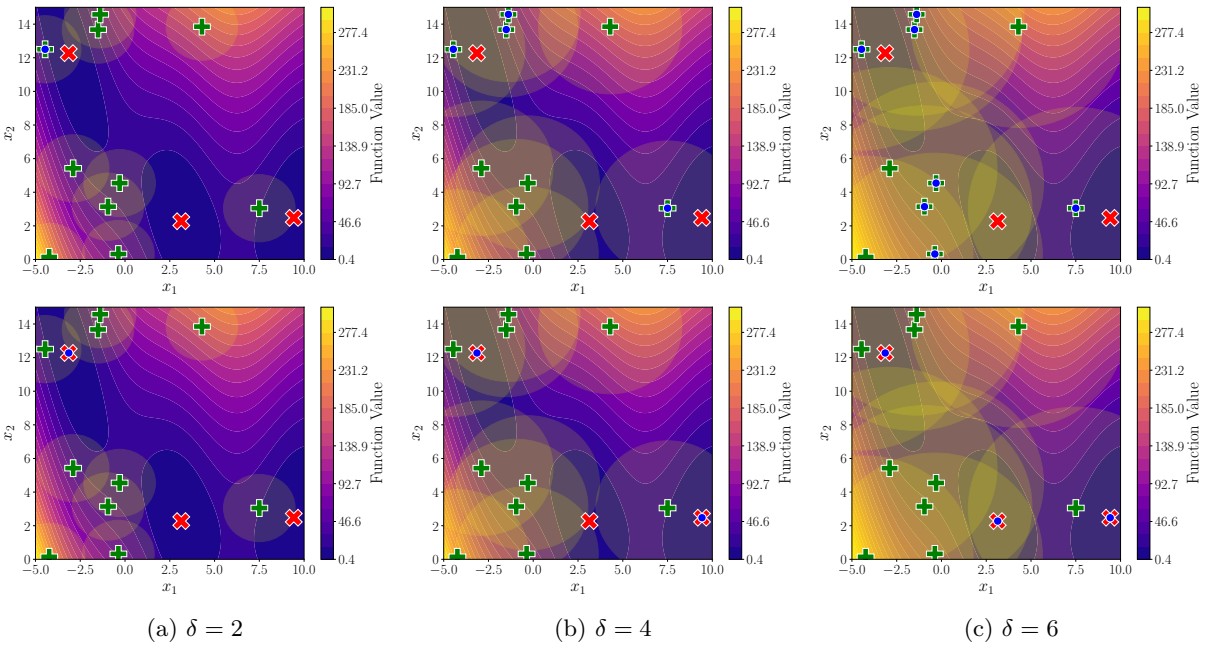

(a) $\delta = 2$          (b) $\delta = 4$          (c) $\delta = 6$

Figure 1: Examples of how precision (upper panels) and recall (lower panels) are computed where *three* global optima (red x) and *ten* query points (green +) are given. Selected points (blue dot) are marked if they are located in the vicinity (yellow circle) of query points. The Branin function is used for these examples and the colorbar visualized in each figure depicts its function values.

## 3 Geometric Metrics for Bayesian Optimization

In this section we introduce new geometric metrics for Bayesian optimization beyond instantaneous, simple, and cumulative regrets. We would like to emphasize that the goal of our new metrics is to complement the conventional regret-based metrics and provide more delicate interpretation of Bayesian optimization algorithms in diverse realistic scenarios.

Suppose that we are given $t$ query points $\mathbf{x}_1, \mathbf{x}_2, \ldots, \mathbf{x}_t$ and $s$ global solutions $\mathbf{x}_1^*, \mathbf{x}_2^*, \ldots, \mathbf{x}_s^*$. An indicator function $\mathbb{I}(z)$ returns 1 if $z$ is true and 0 otherwise.

### 3.1 Precision

We first define precision over $\mathbf{x}_1, \mathbf{x}_2, \ldots, \mathbf{x}_t$ and $\mathbf{x}_1^*, \mathbf{x}_2^*, \ldots, \mathbf{x}_s^*$:

$$\text{precision}_\delta(\{\mathbf{x}_1, \mathbf{x}_2, \ldots, \mathbf{x}_t\}, \{\mathbf{x}_1^*, \mathbf{x}_2^*, \ldots, \mathbf{x}_s^*\}) = \frac{\sum_{i=1}^{t} \vee_{j=1}^{s} \mathbb{I}(\|\mathbf{x}_i - \mathbf{x}_j^*\|_2^2 \le \delta^2)}{t}, \tag{5}$$

where $\delta$ is the radius of ball query and $\vee$ stands for a logical disjunction operation. The goal of this metric is to count the number of query points in the vicinity of global optima. If a metric value is close to one, most query points are located in the close region of global optima. In the spirit of Bayesian optimization, exploration-focused decisions may decrease the values of this metric. Nevertheless, after evaluating a sufficient number of query points, the metric values should increase if a Bayesian optimization algorithm appropriately works.

### 3.2 Recall

Recall over $\mathbf{x}_1, \mathbf{x}_2, \ldots, \mathbf{x}_t$ and $\mathbf{x}_1^*, \mathbf{x}_2^*, \ldots, \mathbf{x}_s^*$ is given by the following:

$$\text{recall}_\delta(\{\mathbf{x}_1, \mathbf{x}_2, \ldots, \mathbf{x}_t\}, \{\mathbf{x}_1^*, \mathbf{x}_2^*, \ldots, \mathbf{x}_s^*\}) = \frac{\sum_{i=1}^{s} \vee_{j=1}^{t} \mathbb{I}(\|\mathbf{x}_i^* - \mathbf{x}_j\|_2^2 \le \delta^2)}{s}, \tag{6}$$

Table 2: Quantitative results on the examples shown in Figures 1 and 10. IR, SR, CR indicate instantaneous, simple, and cumulative regrets, respectively. Note that Figure 10 is presented in Section A

| Example | IR | SR | CR | Precision | | | Recall | | | Average Degree | | | Average Distance | | |
|---|---|---|---|---|---|---|---|---|---|---|---|---|---|---|---|
| | | | | $\delta=2$ | $\delta=4$ | $\delta=6$ | $\delta=2$ | $\delta=4$ | $\delta=6$ | $\delta=2$ | $\delta=4$ | $\delta=6$ | $k=2$ | $k=4$ | $k=6$ |
| Figure 1 | 15.892 | 15.892 | 657.097 | 0.100 | 0.400 | 0.700 | 0.333 | 0.667 | 1.000 | 0.400 | 1.600 | 3.000 | 3.646 | 5.163 | 6.818 |
| Figure 10 | 166.061 | 3.963 | 1278.801 | 0.200 | 0.350 | 0.750 | 0.667 | 0.667 | 1.000 | 1.000 | 4.700 | 8.200 | 2.114 | 2.790 | 3.347 |

where $\delta$ is the radius of ball query and $\vee$ stands for a logical disjunction operation. Its goal is to count the number of global optima in the vicinity of query points. If a metric value is one, all global optima are located in close proximity to some query points. The values of this metric should become closer to one, if a Bayesian optimization method successfully finds all global solutions.

### 3.3 Average Degree

Average degree over $\mathbf{x}_1, \mathbf{x}_2, \ldots, \mathbf{x}_t$ computes the average of degrees over all query points:

$$\text{average-degree}_\delta(\{\mathbf{x}_1, \mathbf{x}_2, \ldots, \mathbf{x}_t\}) = \frac{\sum_{i=1}^{t}\sum_{j=1}^{t}\mathbb{I}(\|\mathbf{x}_i - \mathbf{x}_j\|_2^2 \leq \delta^2) - 1}{t}, \quad (7)$$

where $\delta$ is the radius of ball query. The goal of this metric is to show how many points are mutually close and eventually express an exploitation degree. For example, the metric value becomes one if all points are located in a ball of radius $\delta$ – it implies that all decisions are exploitation-focused.

### 3.4 Average Distance

Average distance over $\mathbf{x}_1, \mathbf{x}_2, \ldots, \mathbf{x}_t$ is defined as the following:

$$\text{average-distance}_k(\{\mathbf{x}_1, \mathbf{x}_2, \ldots, \mathbf{x}_t\}) = \frac{\sum_{i=1}^{t}\sum_{j=1}^{t}\|\mathbf{x}_i - \mathbf{x}_j\|_2 \mathbb{I}(\mathbf{x}_j \in \text{NN}_k(\mathbf{x}_i))}{kt}, \quad (8)$$

where $\text{NN}_k(\mathbf{x}_i)$ is a $k$-nearest neighbor function to obtain $k$ nearest neighbors based on the Euclidean distance. It is to calculate how query points are distributed. If metric values are larger, query points are well-distributed. As an extreme case with four query points in a two-dimensional search space, if all four query points are located on the respective corners of the search space, it will be the maximum value. Thus, this metric indicates the degree of exploration.

As presented in Figures 1 and 10 and Table 2, we show two examples of four geometric metrics as well as three conventional metrics for Bayesian optimization. The Branin function, which is employed in these examples, has three global optimum (red x), and its search space is $[[-5, 10], [0, 15]]$. By using a fixed number of random points (green +), i.e., 10 for Figure 1 and 20 for Figure 10, metric values are reported in Table 2. As expected, our proposed geometric metrics are capable of considering geometric relationships between query points and global optima, while instantaneous, simple, and cumulative regrets only take into account the function evaluations of query points. However, as shown in the results with $\delta = 2, 4, 6$ or $k = 2, 4, 6$, the metric values are sensitive to $\delta$ or $k$. To resolve such an issue, we introduce the parameter-free forms of our metrics in Section 3.5.

### 3.5 Parameter-Free Forms of New Metrics

To propose new metrics without additional parameters, which are hard to determine, we define the parameter-free forms of four metrics, i.e., precision, recall, average degree, and average distance. In general, we integrate out $\delta$ or $k$ by sampling it from a particular distribution. Firstly, we describe parameter-free precision and

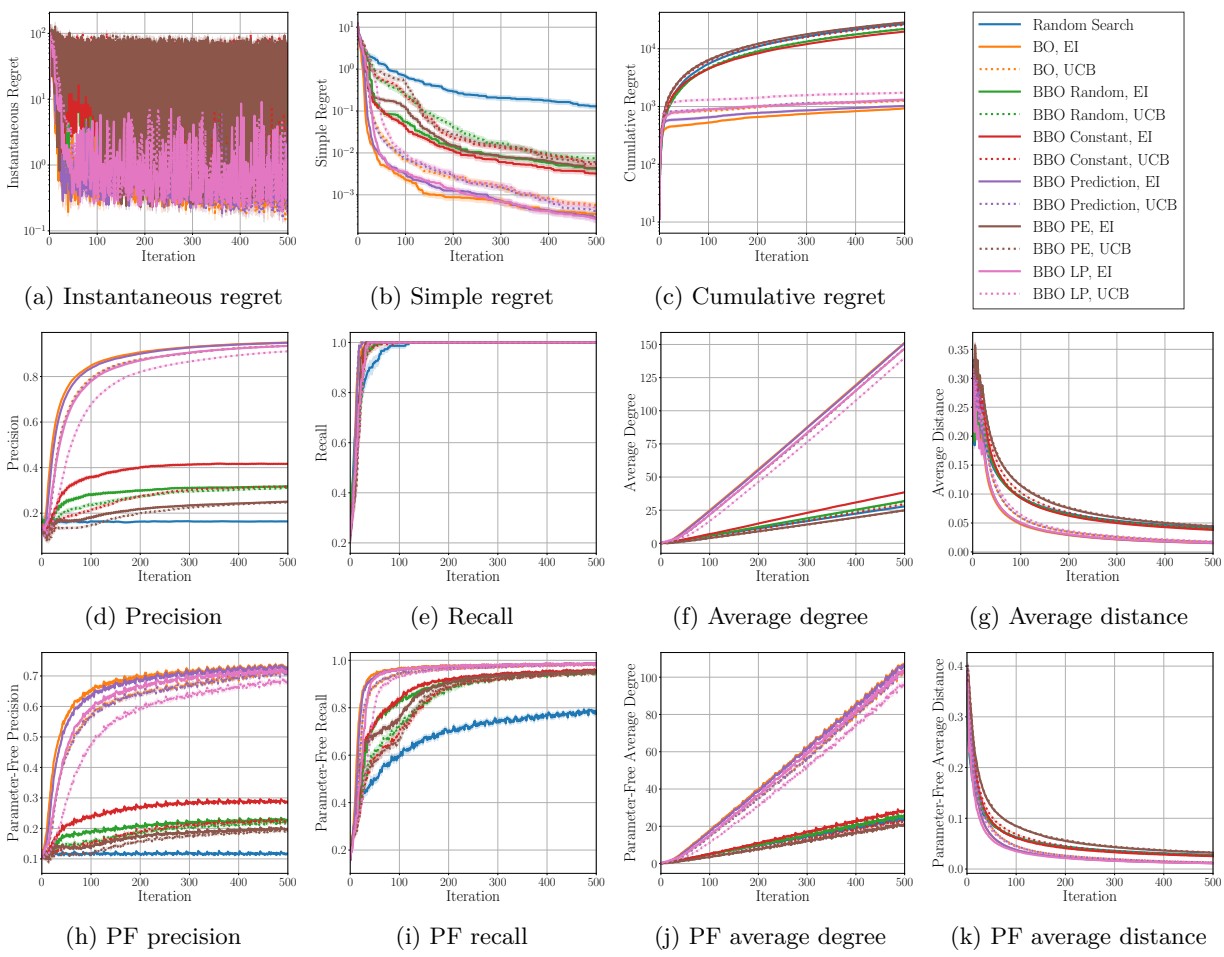

(a) Instantaneous regret     (b) Simple regret     (c) Cumulative regret

(d) Precision     (e) Recall     (f) Average degree     (g) Average distance

(h) PF precision     (i) PF recall     (j) PF average degree     (k) PF average distance

Figure 2: Bayesian optimization results versus iterations for the Branin function. Sample means over 50 rounds and the standard errors of the sample mean over 50 rounds are depicted. PF stands for parameter-free.

parameter-free recall:

$$\text{precision}(\{\mathbf{x}_1, \mathbf{x}_2, \ldots, \mathbf{x}_t\}, \{\mathbf{x}_1^*, \mathbf{x}_2^*, \ldots, \mathbf{x}_s^*\}) = \mathbb{E}_\delta \left[ \text{precision}_\delta(\{\mathbf{x}_1, \mathbf{x}_2, \ldots, \mathbf{x}_t\}, \{\mathbf{x}_1^*, \mathbf{x}_2^*, \ldots, \mathbf{x}_s^*\}) \right]$$

$$= \frac{1}{M} \sum_{i=1}^{M} \text{precision}_{\delta_i}(\{\mathbf{x}_1, \mathbf{x}_2, \ldots, \mathbf{x}_t\}, \{\mathbf{x}_1^*, \mathbf{x}_2^*, \ldots, \mathbf{x}_s^*\}), \quad (9)$$

$$\text{recall}(\{\mathbf{x}_1, \mathbf{x}_2, \ldots, \mathbf{x}_t\}, \{\mathbf{x}_1^*, \mathbf{x}_2^*, \ldots, \mathbf{x}_s^*\}) = \mathbb{E}_\delta \left[ \text{recall}_\delta(\{\mathbf{x}_1, \mathbf{x}_2, \ldots, \mathbf{x}_t\}, \{\mathbf{x}_1^*, \mathbf{x}_2^*, \ldots, \mathbf{x}_s^*\}) \right]$$

$$= \frac{1}{M} \sum_{i=1}^{M} \text{recall}_{\delta_i}(\{\mathbf{x}_1, \mathbf{x}_2, \ldots, \mathbf{x}_t\}, \{\mathbf{x}_1^*, \mathbf{x}_2^*, \ldots, \mathbf{x}_s^*\}), \quad (10)$$

where every $\delta$ is sampled from the exponential distribution with a rate parameter $1/d$: $\delta_i \sim$ Exponential($\delta; 1/d$) and $M$ is the number of samples. Since the exponential distribution, which is defined with $\delta > 0$, is monotonically decreasing as $\delta$ increases, it is suitable for our case. Specifically, the use of this distribution for sampling $M$ parameters naturally considers penalizing a farther region from a particular point. Moreover, we provide such a rate parameter, in order to make the variance of the distribution larger if the dimensionality of an optimization problem is larger. Similarly, parameter-free average degree is also

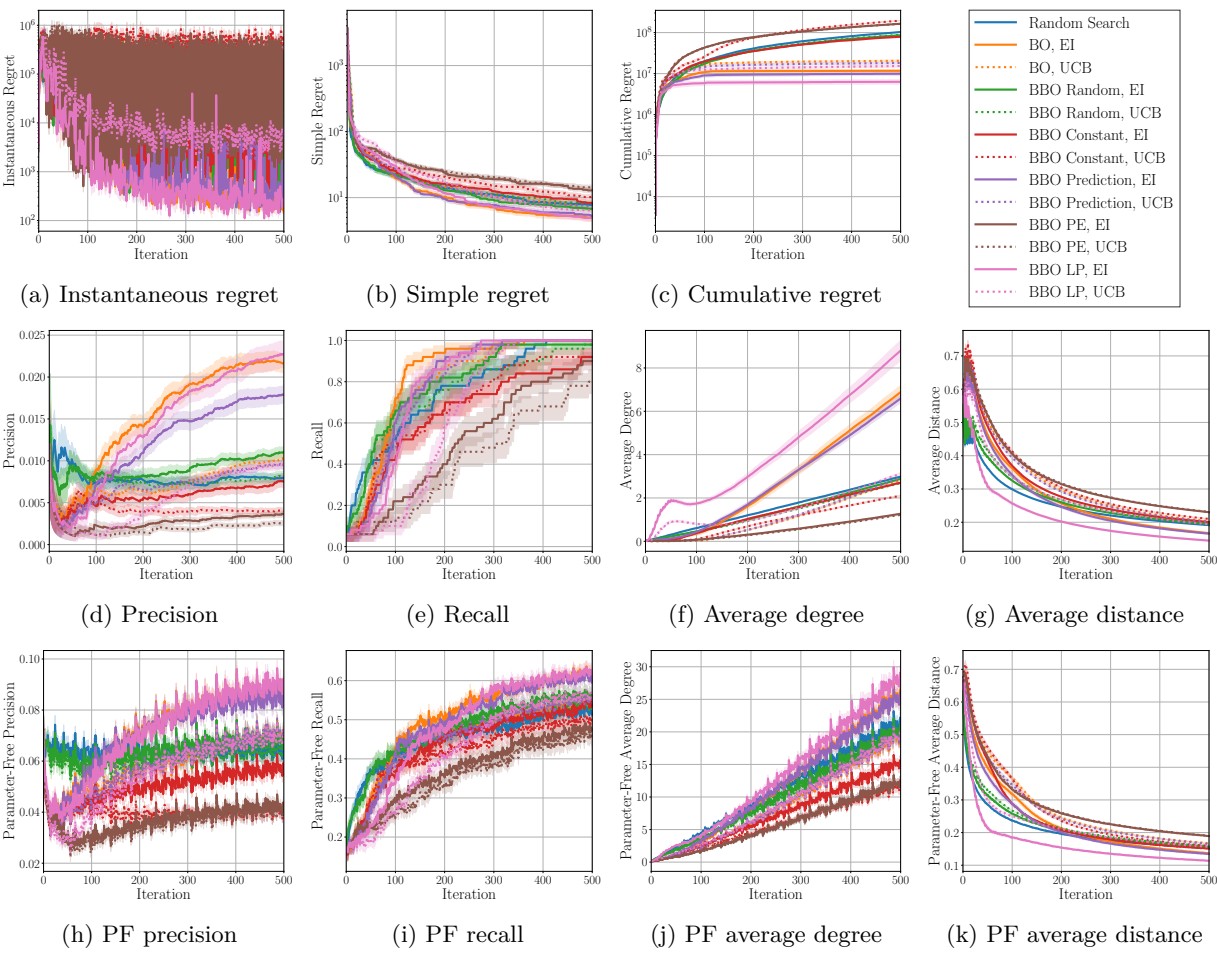

Figure 3: Bayesian optimization results versus iterations for the Zakharov 4D function. Sample means over 50 rounds and the standard errors of the sample mean over 50 rounds are depicted. PF stands for parameter-free.

defined:

$$\text{average-degree}(\{\mathbf{x}_1, \mathbf{x}_2, \ldots, \mathbf{x}_t\}) = \mathbb{E}_\delta \left[ \text{average-degree}_\delta (\{\mathbf{x}_1, \mathbf{x}_2, \ldots, \mathbf{x}_t\}) \right]$$

$$= \frac{1}{M} \sum_{i=1}^{M} \text{average-degree}_{\delta_i} (\{\mathbf{x}_1, \mathbf{x}_2, \ldots, \mathbf{x}_t\}), \tag{11}$$

where $\delta_1, \delta_2, \ldots, \delta_M \sim \text{Exponential}(\delta; 1/d)$. Moreover, parameter-free average distance is defined:

$$\text{average-distance}(\{\mathbf{x}_1, \mathbf{x}_2, \ldots, \mathbf{x}_t\}) = \mathbb{E}_\delta \left[ \text{average-distance}_k (\{\mathbf{x}_1, \mathbf{x}_2, \ldots, \mathbf{x}_t\}) \right]$$

$$= \frac{1}{M} \sum_{i=1}^{M} \text{average-distance}_{k_i} (\{\mathbf{x}_1, \mathbf{x}_2, \ldots, \mathbf{x}_t\}), \tag{12}$$

where every $k$ is sampled from the geometric distribution with a success rate 0.5: $k_i \sim \text{Geometric}(k; 0.5)$ for $i \in [M]$, and $M$ is the number of samples. The choice of the geometric distribution also resorts to its decaying property, where the geometric distribution is essentially the discrete version of the exponential distribution. To sample the number of neighbor neighbors not depending on any factors in a Bayesian optimization process, we set a success probability as 0.5.

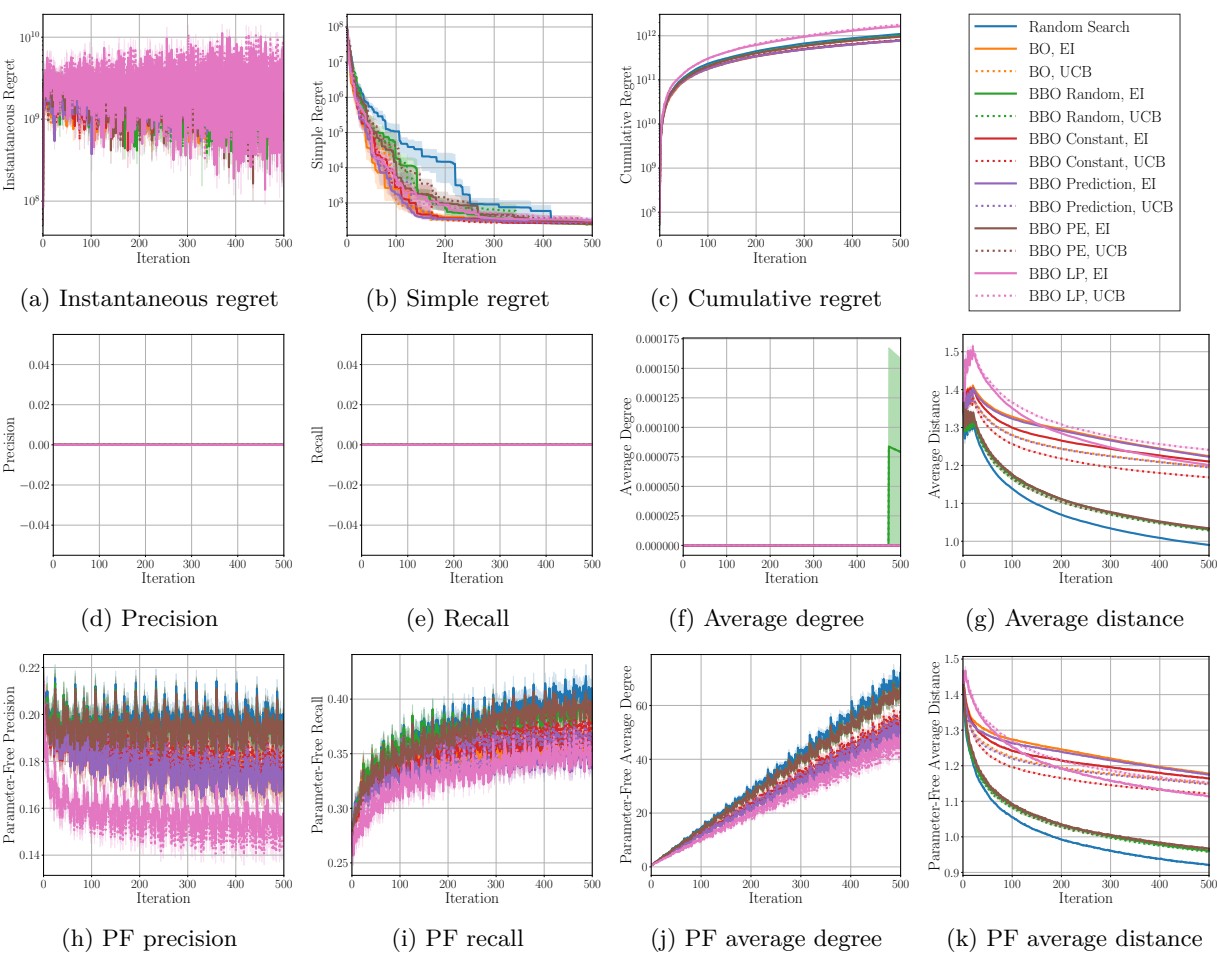

Figure 4: Bayesian optimization results versus iterations for the Zakharov 16D function. Sample means over 50 rounds and the standard errors of the sample mean over 50 rounds are depicted. PF stands for parameter-free.

## 4 Numerical Analysis

To exhibit the characteristics of our proposed metrics, we conduct numerical analysis on our metrics in diverse popular benchmark functions. This analysis is designed to thoroughly understand our geometric metrics as well as the conventional regret-based metrics. In particular, we will visualize metric values over iterations for different metrics, and then show the Spearman's rank correlation coefficients between metrics, based on the results of the metric values over iterations. For the geometric metrics defined with $\delta$, $\delta = \text{max-dist}(\mathcal{X})/10$ where $\text{max-dist}(\mathcal{X})$ is the maximum distance of a search space $\mathcal{X}$. If $\mathcal{X} = [[0, 10], [-5, 10], [-10, 10]]$, $\text{max-dist}(\mathcal{X}) = \sqrt{725}$. In this numerical analysis, we normalize all search spaces to $[0, 1]^d$ in order to fairly compare diverse benchmarks and various algorithms. Thus, $\text{max-dist}(\mathcal{X}) = \sqrt{d}$ in these cases. In addition, we use $k = \min(\max(t/5, 1), 5)$ for the average distance. For our parameter-free metrics, $M = 100$. Moreover, we evaluate 500 points for each algorithm where 5 initial points sampled from the uniform distribution are given, and repeat each algorithm 50 times for all experiments. For batch algorithms, a batch size is set as 5. Consequently, 100 batches are sequentially selected so that 500 points are acquired.

**Baseline Methods.** Surrogate models of all methods are implemented with Gaussian processes with Matérn 5/2 kernels (Rasmussen & Williams, 2006) and we use either expected improvement (Jones et al., 1998) or Gaussian process upper confidence bound (Srinivas et al., 2010) for the acquisition function of each

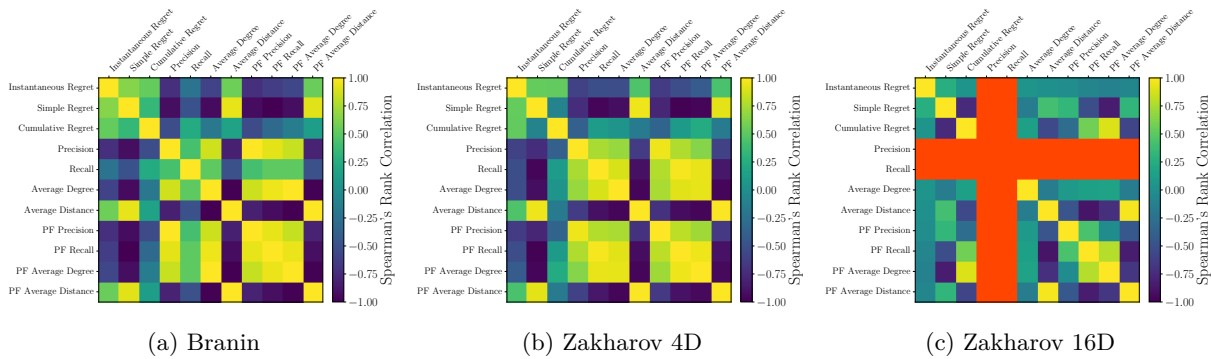

Figure 5: Spearman's rank correlation coefficients between metrics for different benchmark functions such as the Branin, Zakharov 4D, and Zakharov 16D functions. Red regions indicate the coefficients with NaN values. More results are depicted in Figures 26 and 27.

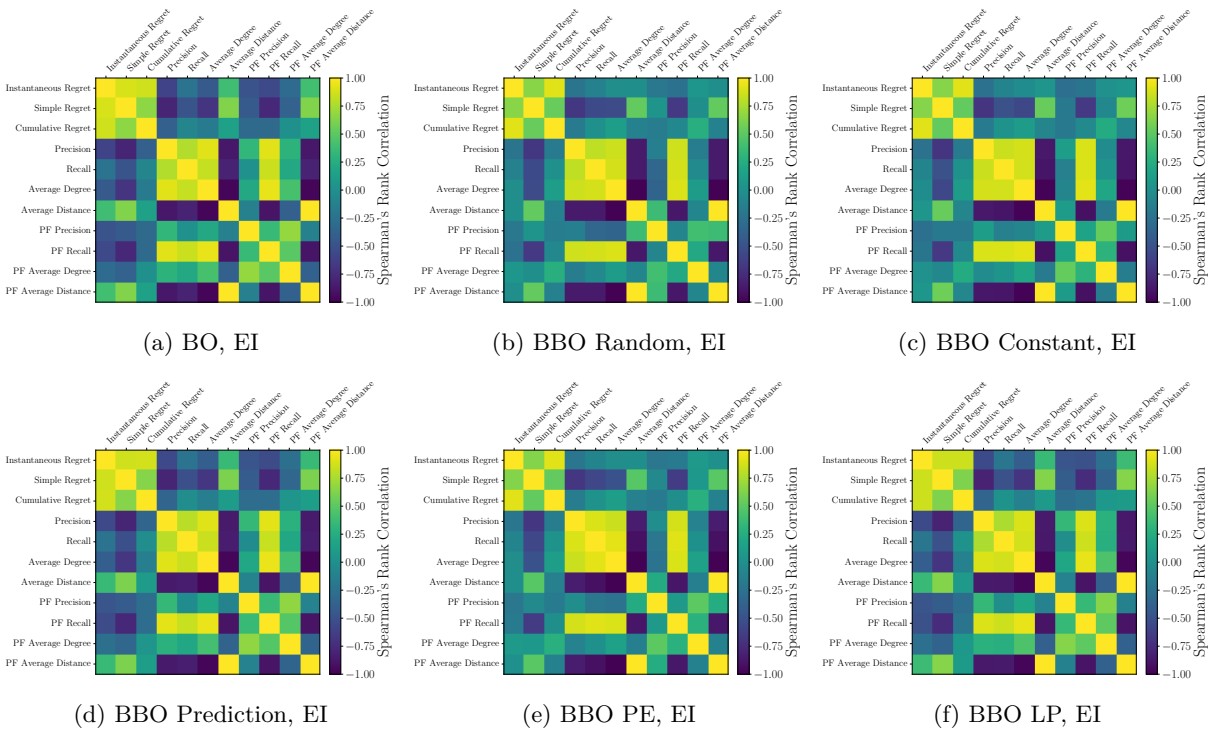

Figure 6: Spearman's rank correlation coefficients between metrics for different Bayesian optimization algorithms with the expected improvement.

baseline method. To find kernel parameters of the Gaussian process regression model, marginal likelihood is maximized using BFGS (Nocedal & Wright, 2006). Moreover, L-BFGS-B (Byrd et al., 1995) is employed in optimizing an acquisition function with 128 initial points. We utilize the following baseline strategies:

(i) Random search: This is a random algorithm to select points from the uniform distribution;

(ii) Bayesian optimization: It is a vanilla Bayesian optimization algorithm, which is denoted as BO;

(iii) Batch Bayesian optimization with random search: This approach, which is denoted as BBO Random, selects a single point using the vanilla Bayesian optimization and the other points using random search in the process of selecting a batch;

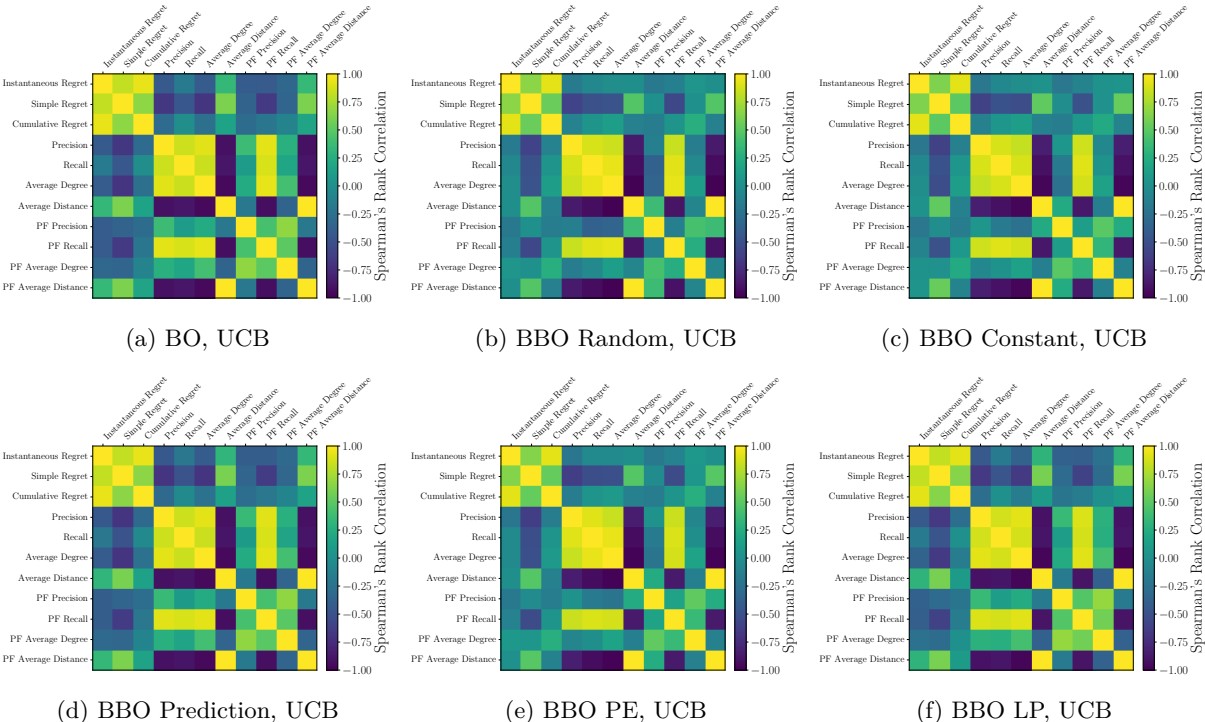

Figure 7: Spearman's rank correlation coefficients between metrics for different Bayesian optimization algorithms with the Gaussian process upper confidence bound.

(iv) Batch Bayesian optimization with a constant (Ginsbourger et al., 2010): It sequentially selects each point in a batch where a fake evaluation with a particular constant is provided every acquisition. After acquiring a batch, query points in the batch are evaluated. The fixed constant is set as 100 and this method is denoted as BBO Constant;

(v) Batch Bayesian optimization with predicted function values (Ginsbourger et al., 2010): This strategy is similar to BBO Constant, but the evaluations of select points are set as the posterior means of Gaussian process regression. It is denoted as BBO Prediction;

(vi) Batch Bayesian optimization with pure exploration (Contal et al., 2013): This method, which is denoted as BBO PE, sequentially chooses points to evaluate using Bayesian optimization with pure exploration, while the first query point is selected by the vanilla Bayesian optimization. Since the posterior variances of Gaussian process regression do not depend on function evaluations, the acquisition strategy of pure exploration can be run without evaluating query points;

(vii) Batch Bayesian optimization with local penalization (González et al., 2016): This algorithm, which is denoted as BBO LP, sequentially determines a batch of query points by penalizing an acquisition function with a local penalization function.

We do not necessitate confining Bayesian optimization methods to batch Bayesian optimization. However, in order to show difference between metrics by utilizing the inherent nature of batch Bayesian optimization methods on the diversification of query points, we adopt them as baseline methods.

## 4.1 Analysis on Metric Values over Iterations

We report the arithmetic means of metric values and their standard errors of the sample mean, as illustrated in Figures 2, 3, and 4 and Figures 11 to 24, which are included in the appendices. Empirical analysis

with a total of 11 metrics is demonstrated by optimizing several benchmark functions through Bayesian optimization. We find that the metric values of instantaneous regret, simple regret, average distance, and parameter-free average distance generally decrease as Bayesian optimization iterates. On the other hand, the metric values of the other metrics such as cumulative regret, precision, recall, average degree, parameter-free precision, parameter-free recall, and parameter-free average degree tend to increase, as iterations proceed. These results are expected because of the goals of the respective metrics.

As shown in Figure 2, although the metric values of the simple regret become closer to zero and the values of the cumulative regret also converge to specific values for some algorithms, the values of the parameter-free recall do not reach to one. It implies that the algorithms used in this analysis fail to find multiple global solutions. Such results are not demonstrated by the use of the conventional regret-based metrics. Also, the metric values of the average degree, average distance, and parameter-free forms of the average degree and average distance for BBO Random and BBO PE are closer to their values for random search, compared to the results of the simple and cumulative regrets. We presume that the components of random search and pure exploration in the respective algorithms make those values more similar to the values for random search. This observation supports our motivation on the development of geometric metrics for measuring the degrees of exploration and exploitation.

Interestingly, the precision, recall, and average degree struggle to measure the performance of Bayesian optimization algorithms in higher-dimensional problems; see the 16-dimensional case shown in Figure 4. To ensure that these metrics work appropriately, we need to carefully select $\delta$ in their definitions. However, this parameter selection task is almost infeasible due to the problem formulation that a black-box function is optimized. In contrast to these metrics involved with an additional parameter, our parameter-free metrics successfully compare distinct optimization algorithms as presented in Figure 4.

## 4.2 Analysis on the Spearman's Rank Correlation Coefficients between Metrics

We investigate the Spearman's rank correlation coefficients between diverse metrics, in order to reveal relationships between metrics. Given the means of metric values over 50 rounds, the Spearman's rank correlation coefficients are calculated:

$$r_s = \frac{\mathrm{cov}(\mathrm{rank}(X), \mathrm{rank}(Y))}{\sigma_{\mathrm{rank}(X)} \sigma_{\mathrm{rank}(Y)}}, \tag{13}$$

where $\mathrm{rank}(X)$ is a function to produce the rank of a given variable $X$, $\mathrm{cov}(X_1, X_2)$ indicates the covariance of two input variables $X_1$, $X_2$, and $\sigma_X$ corresponds to the standard deviation of a given variable $X$.

We conduct two empirical analyses using the Spearman's rank correlation coefficients: the coefficients between metrics for each benchmark function and for each optimization algorithm. As shown in Figure 5 and Figures 26 and 27 of the appendices, the simple regret is highly correlated with the average distance and parameter-free average distance

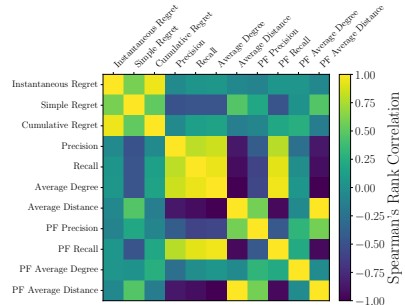

Figure 8: Spearman's rank correlation coefficients between metrics for the random search method.

in general. Moreover, the average degree, parameter-free precision, parameter-free recall, and parameter-free average degree are positively correlated with each other. On the contrary, the average distance and parameter-free average distance are negatively correlated with the average degree, parameter-free precision, parameter-free recall, and parameter-free average degree. In addition, the coefficients with NaN values appear in higher-dimensional cases, since metric values are all identical; see the results of the precision and recall in 16-dimensional examples. These results can be thought of as the evidence of the need to propose parameter-free metrics, instead of selecting an additional parameter carefully.

As presented in Figures 6, 7, and 8, the precision, recall, and average degree are positively correlated with each other, while they are negatively correlated with the average distance and its parameter-free form. In this analysis, the relationships between instantaneous, simple, and cumulative regrets are clearly affirmed. Specifically, the instantaneous and cumulative regrets are highly correlated with each other, and they are loosely correlated with the simple regret.

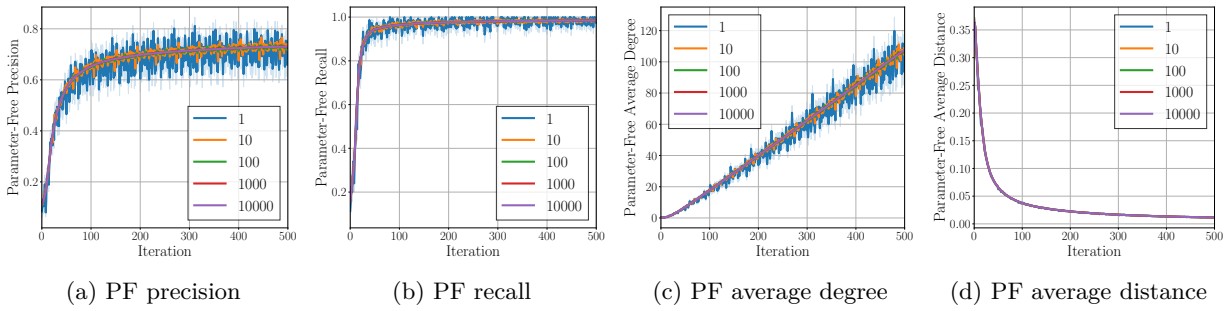

| (a) PF precision | (b) PF recall | (c) PF average degree | (d) PF average distance |

Figure 9: Impact of the number of samples $M$ for parameter-free metrics. The Branin function is optimized using the vanilla Bayesian optimization with the expected improvement.

## 5 Discussion and Limitations on Our Geometric Metrics

**Related Work.** Wessing & Preuss (2017) have been proposed alternative metrics such as averaged Hausdorff distance and peak ratio. These metrics can be used to evaluate multi-local optimization and global optimization. On the other hand, we can directly utilize the information from the posterior predictive distributions of a surrogate model to measure the degrees of exploration and exploitation. For every decision through Bayesian optimization, we can calculate the contributions of function value and variance estimates. However, the geometry of the history of query points are not counted for these calculations.

**On Parameter-Free Metrics.** One drawback of our parameter-free metrics is that their values are prone to oscillating. Since the calculation of the parameter-free metrics include a step for sampling $M$ parameters, this consequence occurs. As shown in Figure 9, a larger $M$ stabilizes metric values, while the use of a larger $M$ makes the calculation slower. Therefore, we need to take into account such a trade-off in the process of selecting $M$. According to these results, $M$ should be equal to or greater than 100.

**On Global Solution Information.** To deal with regret-based metrics or our metrics excluding average degree, average distance, and their parameter-free forms, we require knowing the function value or locations of global solutions; see Table 1 for the detailed comparisons. However, it is rarely possible to obtain this information in real-world problems. We leave such a topic on new metrics without global solution information for future work. As a naïve future direction, we can define the approximation of global solutions utilizing the points that have been acquired. It is noteworthy that our average degree, average distance, parameter-free average degree, and parameter-free average distance are defined with the locations of query points only.

## 6 Conclusion

We have proposed new geometric metrics that can be used to measure the performance of Bayesian optimization. By utilizing information on the geometry of acquired points and potentially global solutions, the precision, recall, average degree, and average distance are defined. Moreover, to mitigate the need to use an additional parameter for the geometric metrics, we also suggested their parameter-free forms. In the end, we provided numerical analysis on metric values over optimization iterations and the Spearman's rank correlation coefficients between various metrics, by testing several Bayesian optimization algorithms as well as random search on a variety of benchmark functions.

### Broader Impact Statement

This work does not have a direct broader impact in that it suggests new metrics for Bayesian optimization. However, from the perspective of the development of optimization algorithms, our work can be used to identify solutions of harmful and unethical optimization tasks. In response to such an issue, we should be aware of how Bayesian optimization algorithms can be applied in unethical tasks when we devise optimization strategies using our geometric metrics.

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

# A   Additional Examples of Precision and Recall for the Branin Function

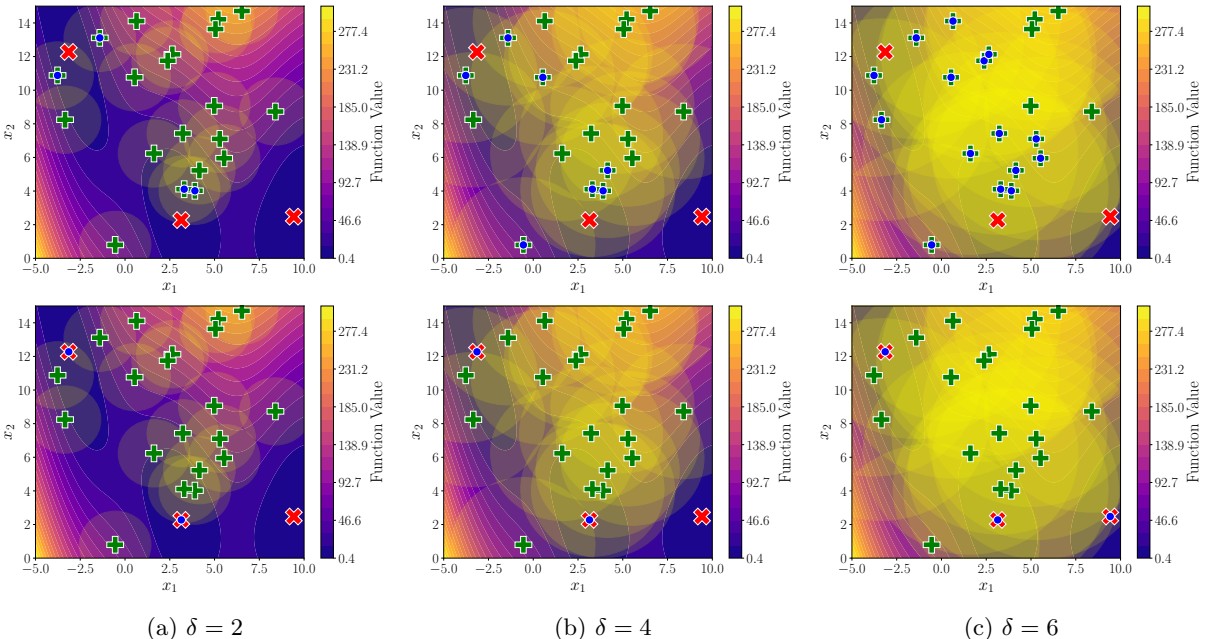

(a) $\delta = 2$            (b) $\delta = 4$            (c) $\delta = 6$

Figure 10: Examples of how precision (upper panels) and recall (lower panels) are computed where *three* global optima (red x) and *twenty* query points (green +) are given. Selected points (blue dot) are marked if they are located in the vicinity (yellow circle) of query points. The Branin function is used for these examples and the colorbar visualized in each figure depicts its function values.

Similar to Figure 1, Figure 10 illustrates the examples of how precision and recall are computed in the case of the Branin function, varying the size of ball query.

# B Additional Analysis on Metric Values over Iterations

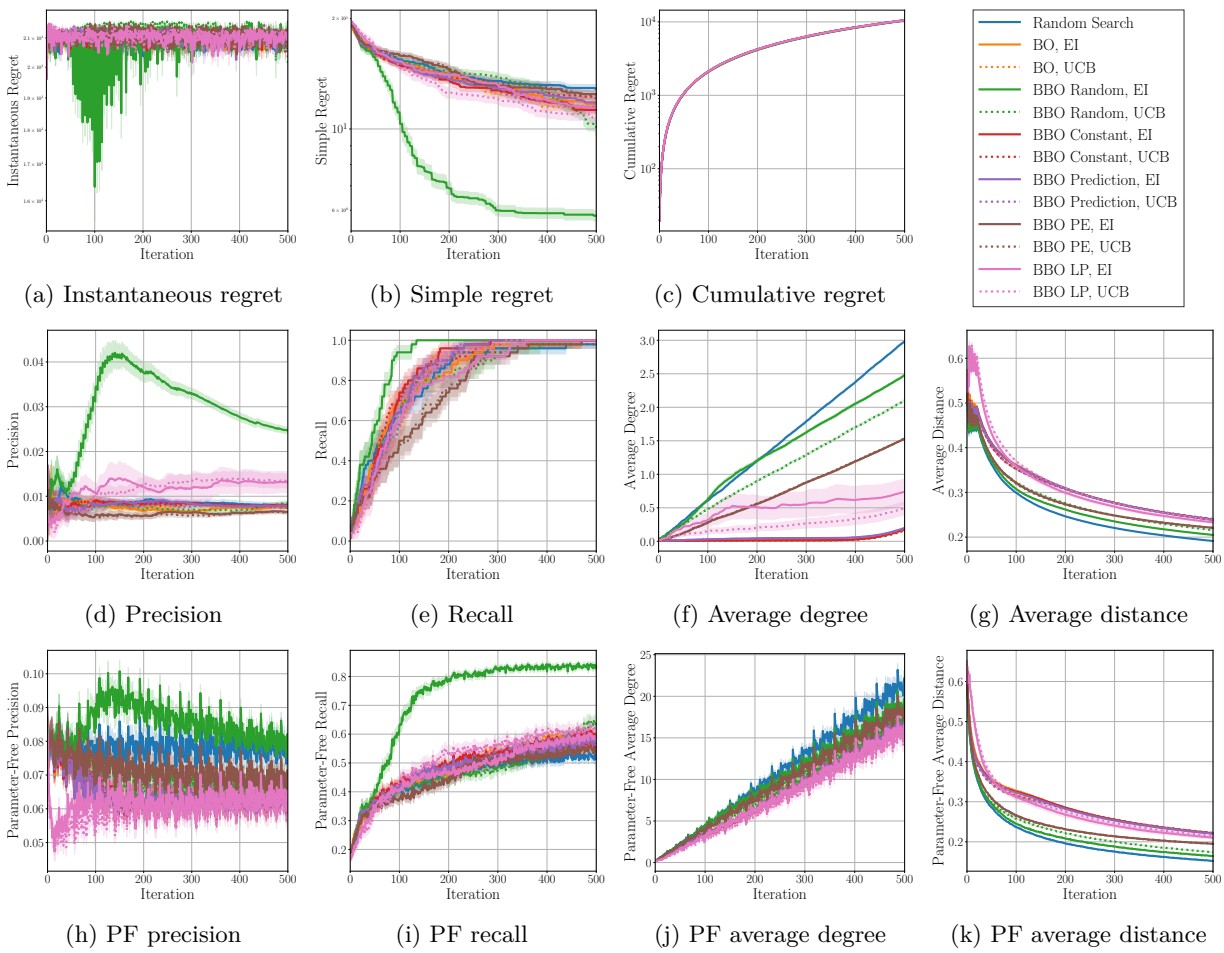

Figure 11: Bayesian optimization results versus iterations for the Ackley 4D function. Sample means over 50 rounds and the standard errors of the sample mean over 50 rounds are depicted. PF stands for parameter-free.

In addition to the results reported in the main article, we present more analysis on metric values over iterations in Figures 11 to 24. The experimental settings of these results are the same as the corresponding settings described in the main article.

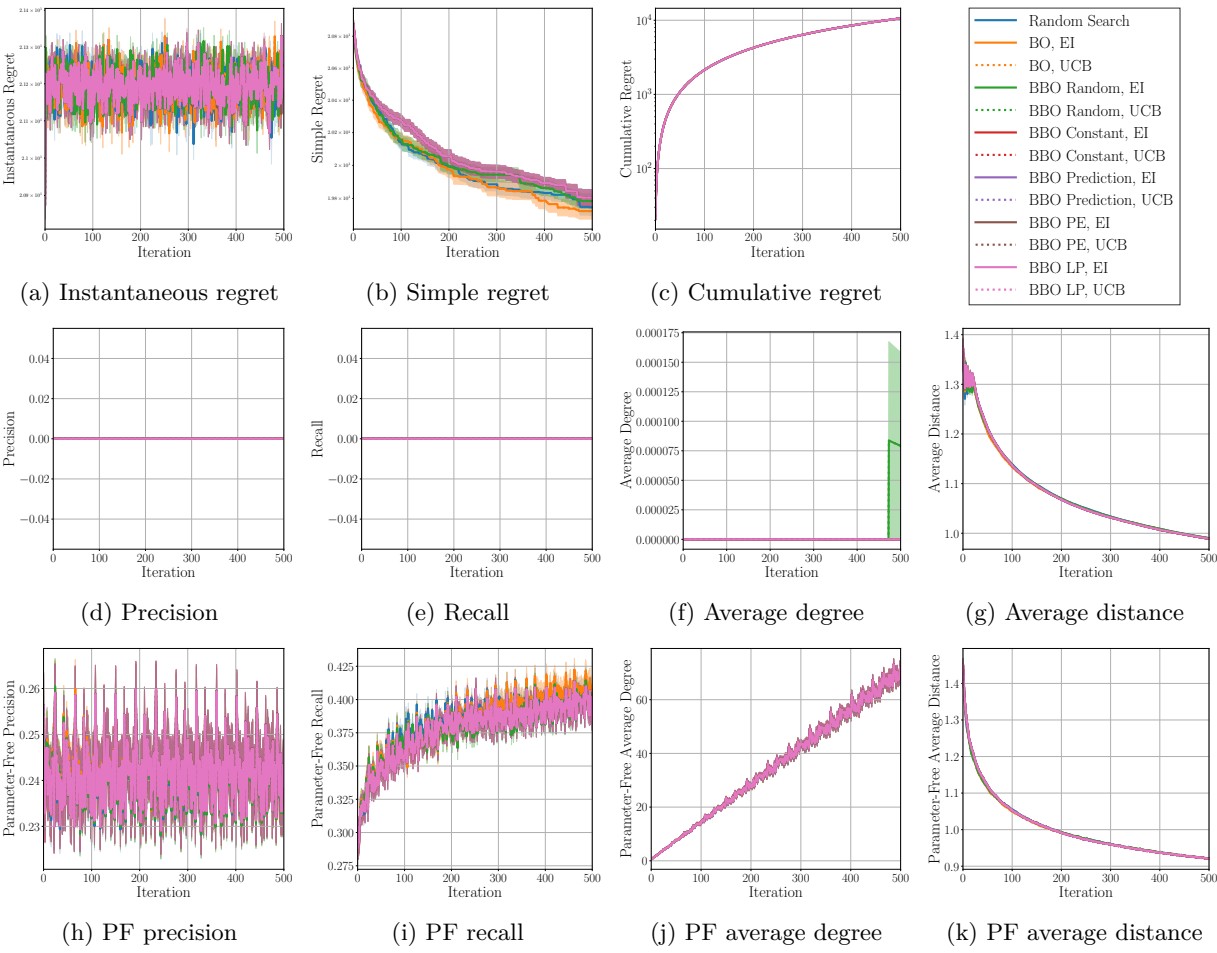

Figure 12: Bayesian optimization results versus iterations for the Ackley 16D function. Sample means over 50 rounds and the standard errors of the sample mean over 50 rounds are depicted. PF stands for parameter-free.

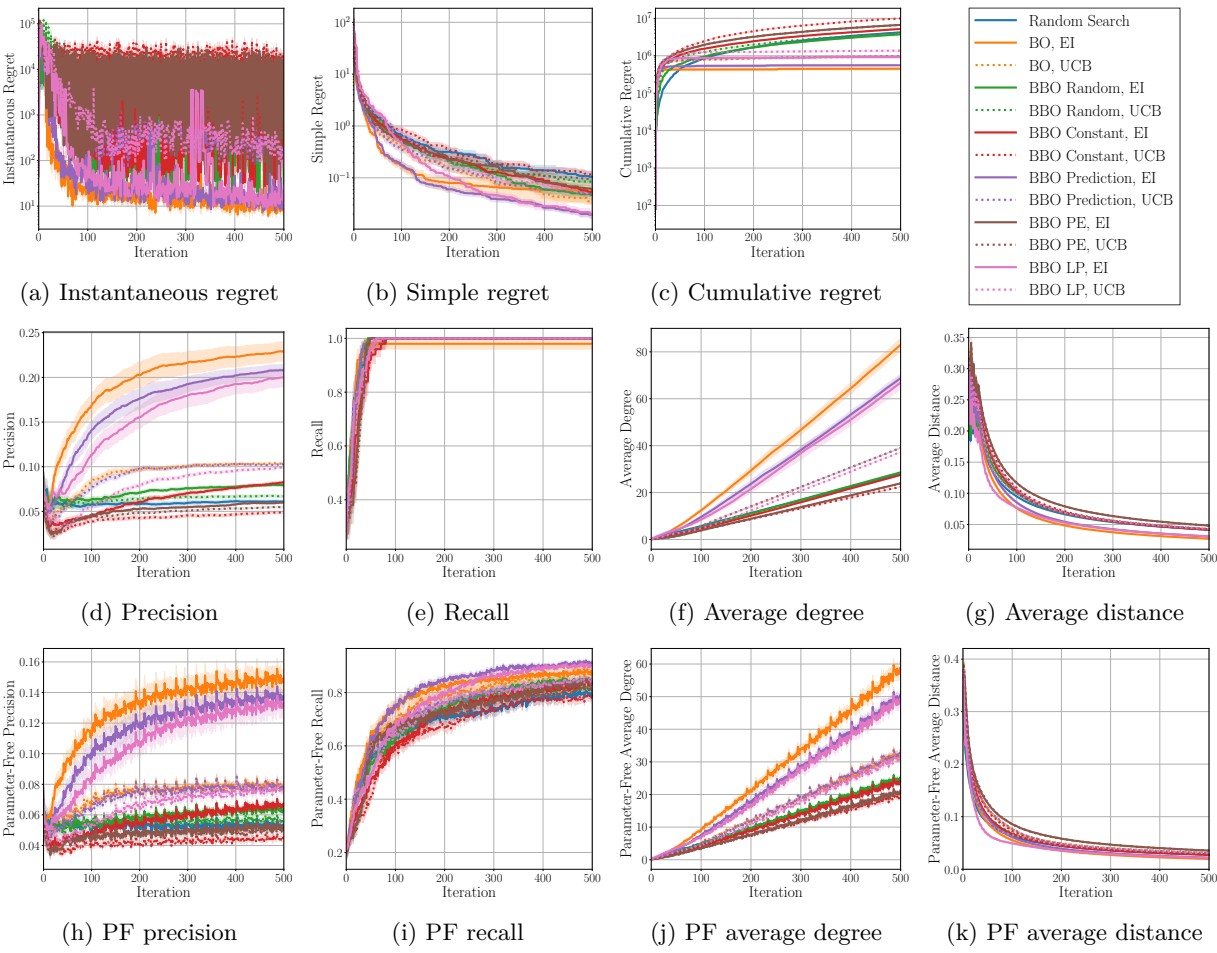

Figure 13: Bayesian optimization results versus iterations for the Beale function. Sample means over 50 rounds and the standard errors of the sample mean over 50 rounds are depicted. PF stands for parameter-free.

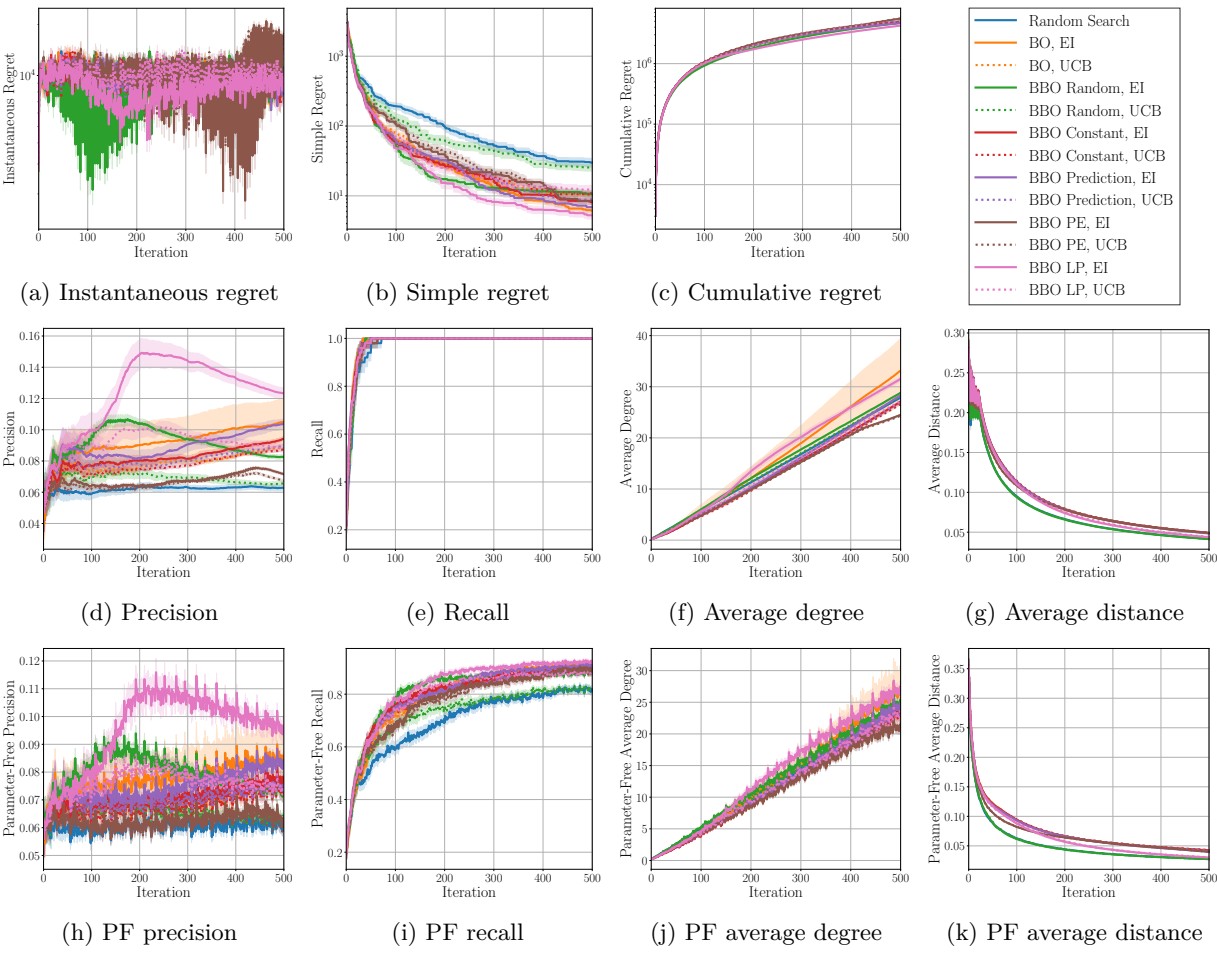

Figure 14: Bayesian optimization results versus iterations for the Bohachevsky function. Sample means over 50 rounds and the standard errors of the sample mean over 50 rounds are depicted. PF stands for parameter-free.

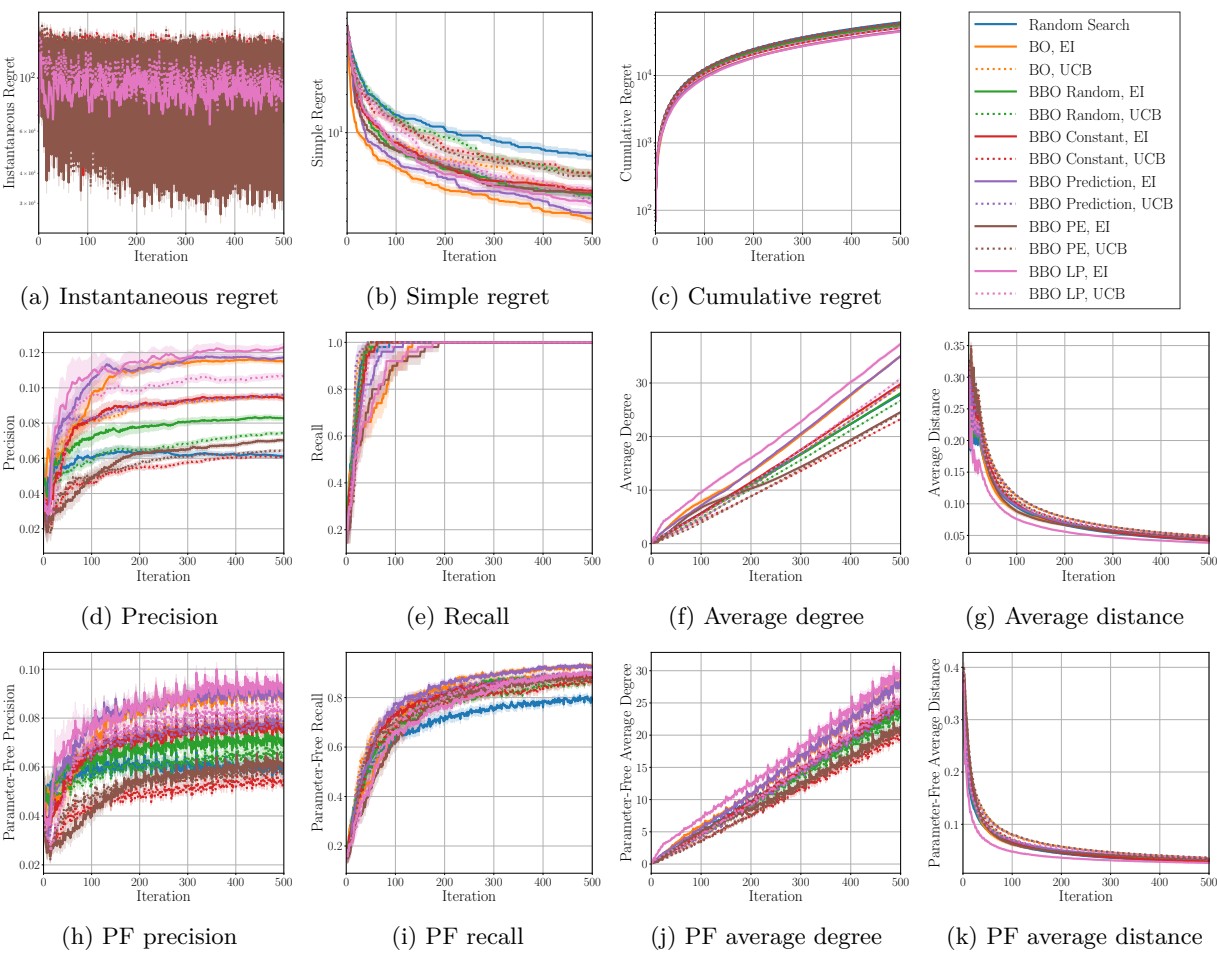

(a) Instantaneous regret     (b) Simple regret     (c) Cumulative regret

(d) Precision     (e) Recall     (f) Average degree     (g) Average distance

(h) PF precision     (i) PF recall     (j) PF average degree     (k) PF average distance

Figure 15: Bayesian optimization results versus iterations for the Bukin 6 function. Sample means over 50 rounds and the standard errors of the sample mean over 50 rounds are depicted. PF stands for parameter-free.

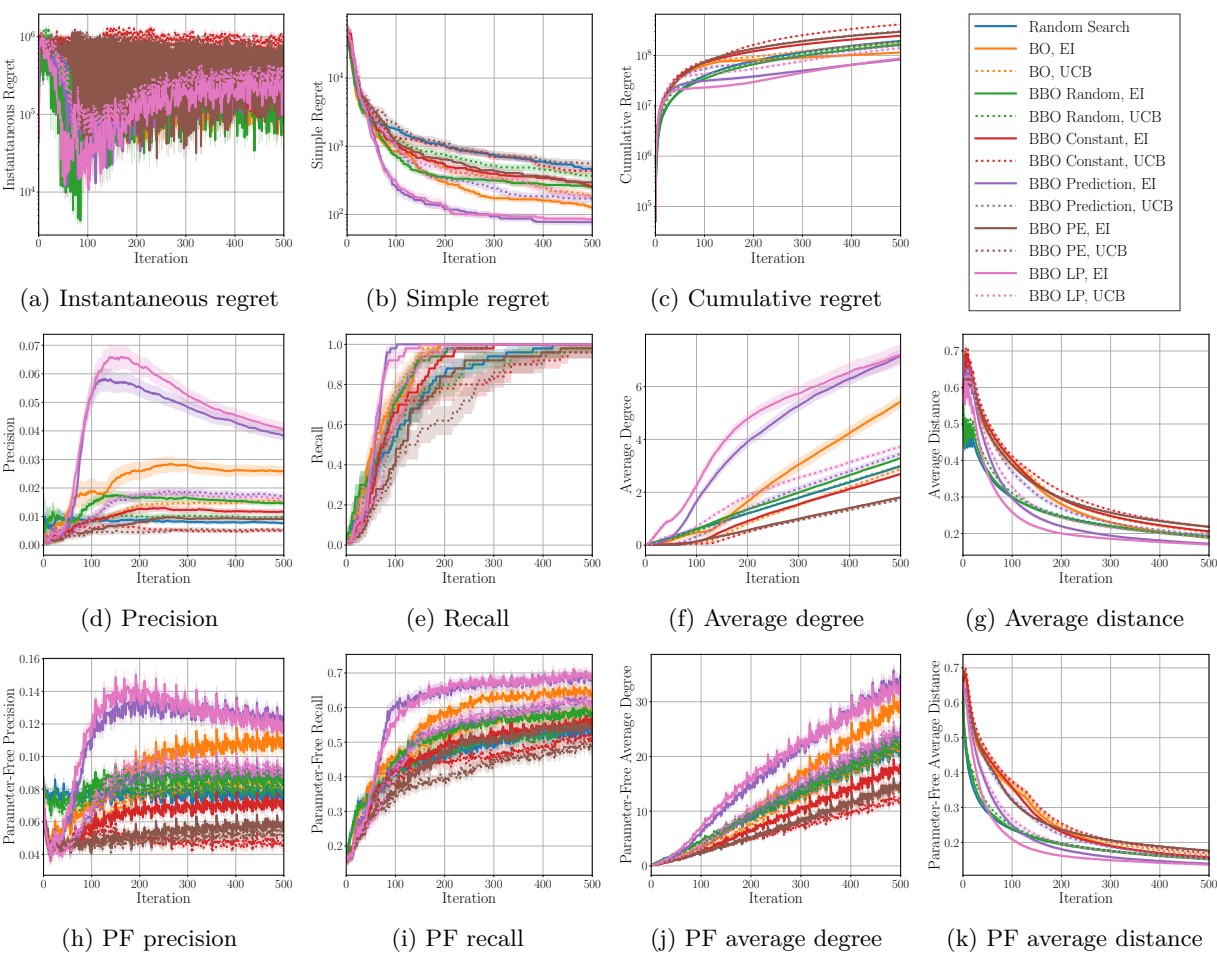

(a) Instantaneous regret     (b) Simple regret     (c) Cumulative regret

(d) Precision     (e) Recall     (f) Average degree     (g) Average distance

(h) PF precision     (i) PF recall     (j) PF average degree     (k) PF average distance

Figure 16: Bayesian optimization results versus iterations for the Colville function. Sample means over 50 rounds and the standard errors of the sample mean over 50 rounds are depicted. PF stands for parameter-free.

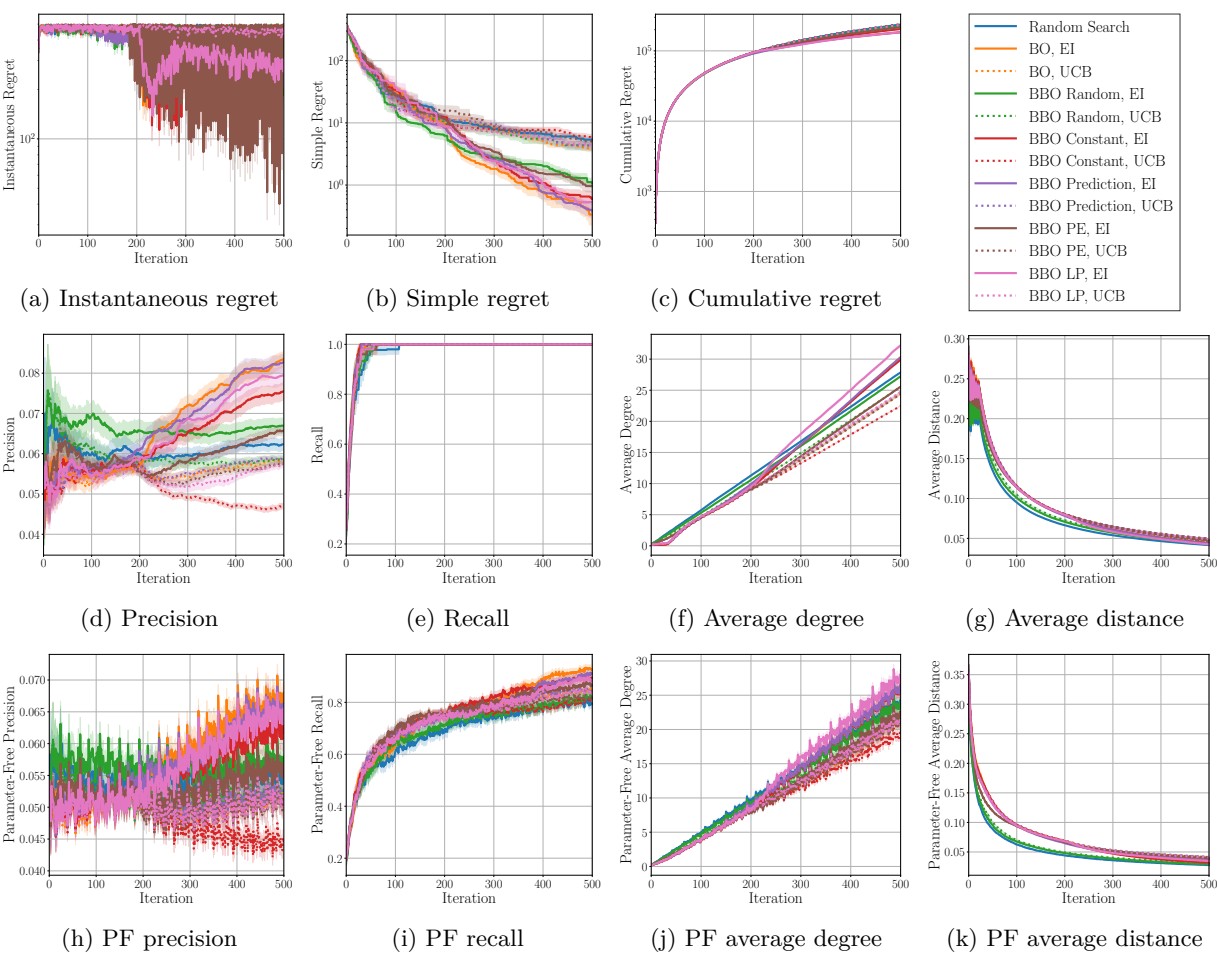

(a) Instantaneous regret     (b) Simple regret     (c) Cumulative regret

(d) Precision     (e) Recall     (f) Average degree     (g) Average distance

(h) PF precision     (i) PF recall     (j) PF average degree     (k) PF average distance

Figure 17: Bayesian optimization results versus iterations for the De Jong 5 function. Sample means over 50 rounds and the standard errors of the sample mean over 50 rounds are depicted. PF stands for parameter-free.

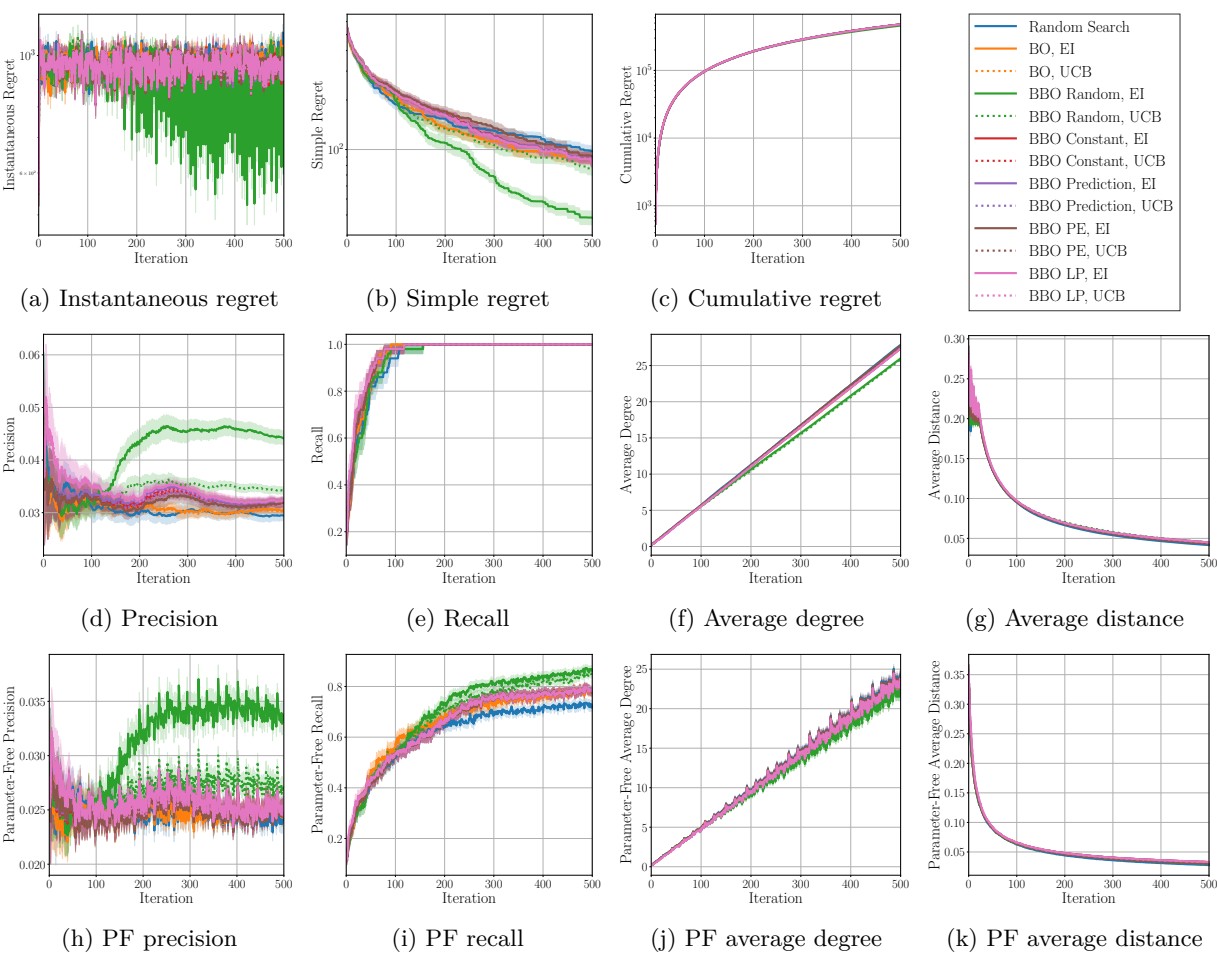

(a) Instantaneous regret      (b) Simple regret      (c) Cumulative regret

(d) Precision      (e) Recall      (f) Average degree      (g) Average distance

(h) PF precision      (i) PF recall      (j) PF average degree      (k) PF average distance

Figure 18: Bayesian optimization results versus iterations for the Eggholder function. Sample means over 50 rounds and the standard errors of the sample mean over 50 rounds are depicted. PF stands for parameter-free.

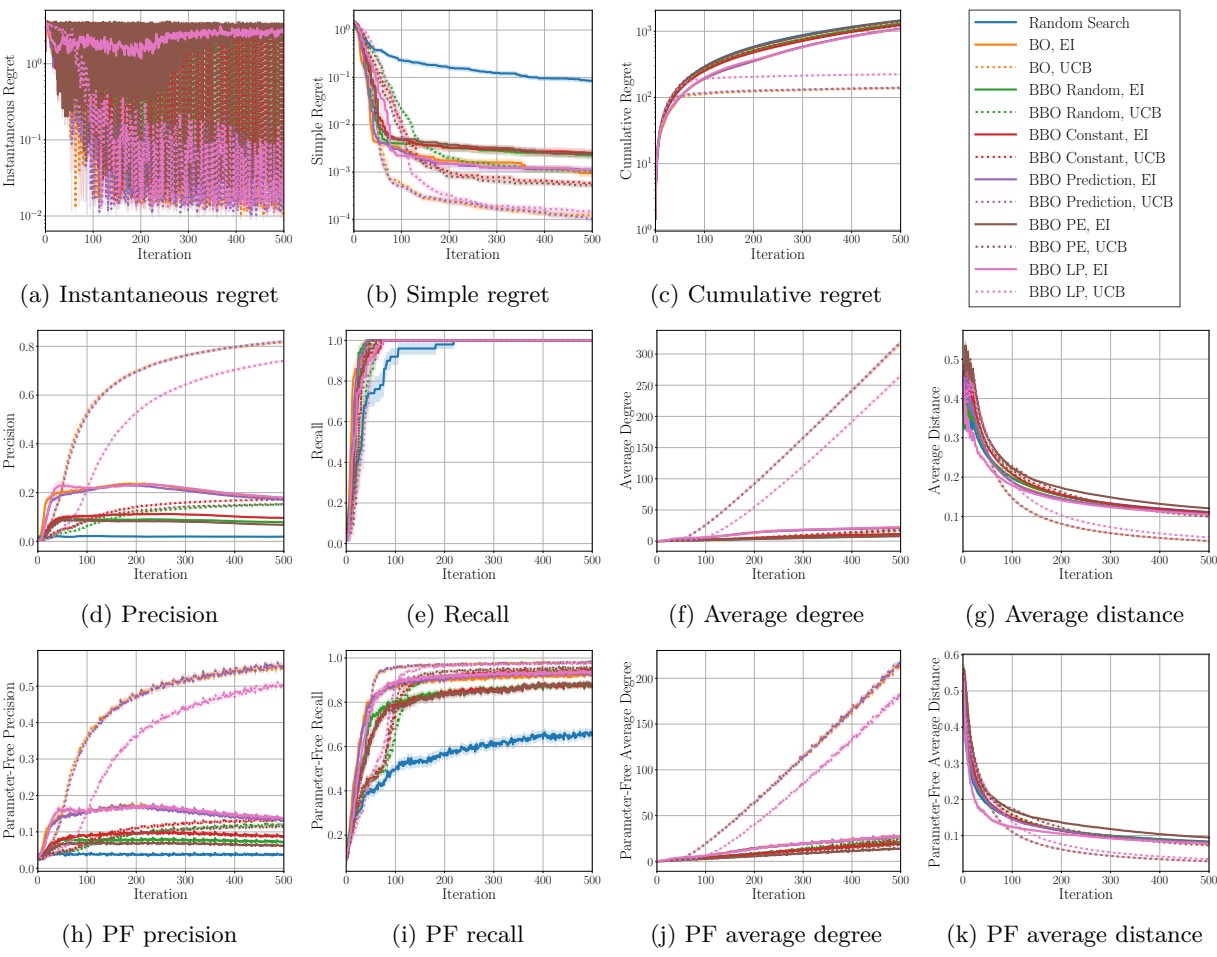

Figure 19: Bayesian optimization results versus iterations for the Hartmann 3D function. Sample means over 50 rounds and the standard errors of the sample mean over 50 rounds are depicted. PF stands for parameter-free.

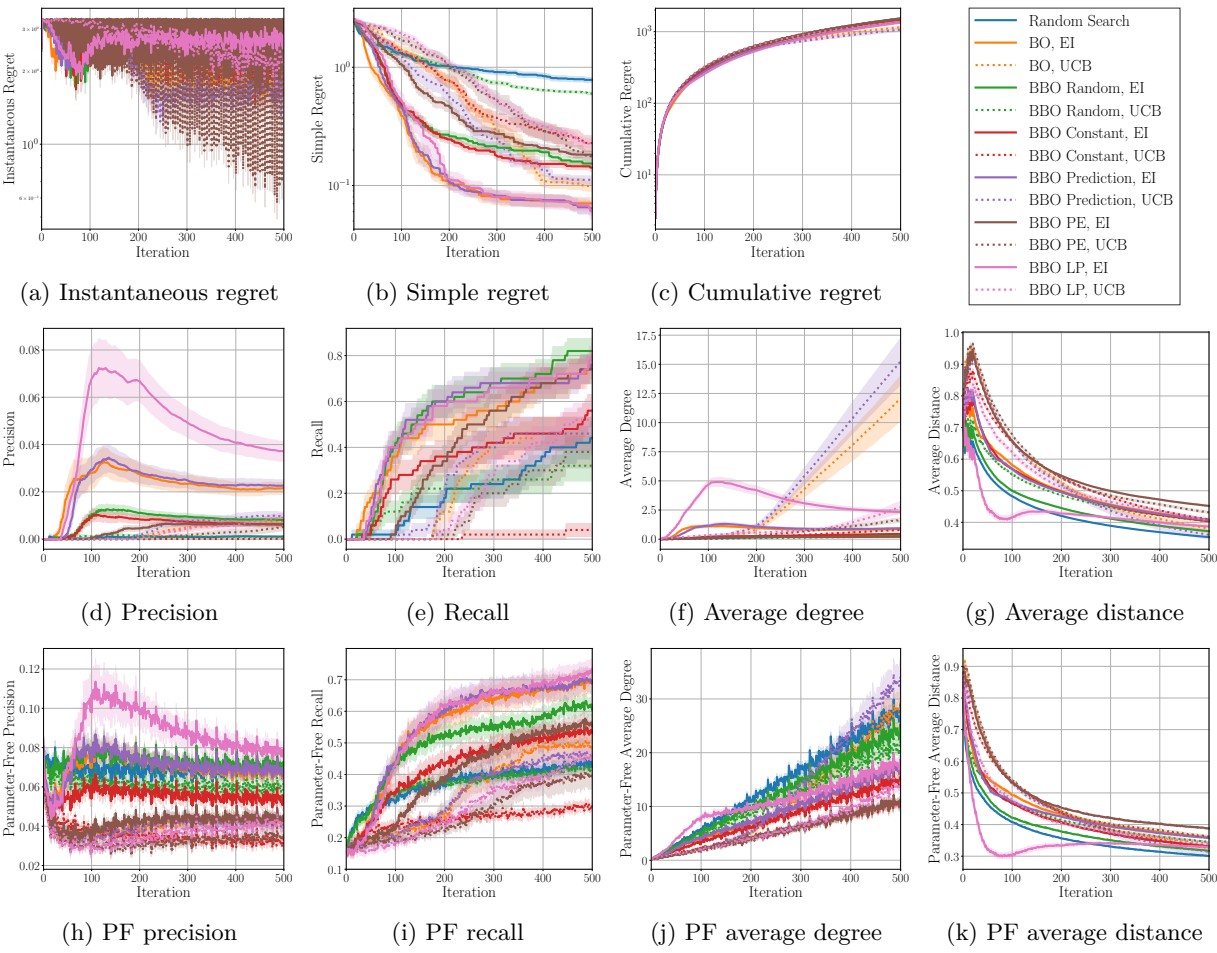

Figure 20: Bayesian optimization results versus iterations for the Hartmann 6D function. Sample means over 50 rounds and the standard errors of the sample mean over 50 rounds are depicted. PF stands for parameter-free.

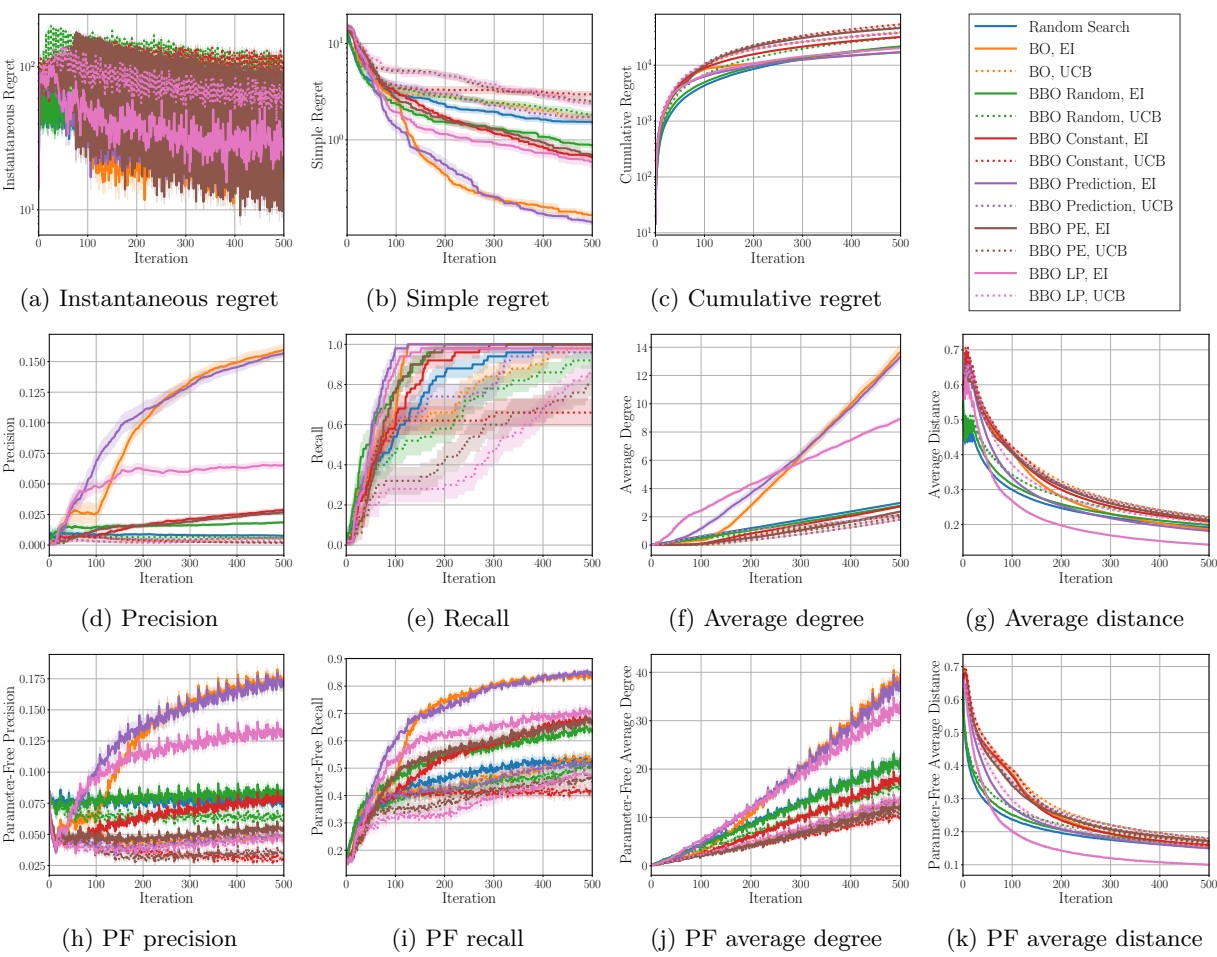

(a) Instantaneous regret   (b) Simple regret   (c) Cumulative regret

(d) Precision   (e) Recall   (f) Average degree   (g) Average distance

(h) PF precision   (i) PF recall   (j) PF average degree   (k) PF average distance

Figure 21: Bayesian optimization results versus iterations for the Levy 4D function. Sample means over 50 rounds and the standard errors of the sample mean over 50 rounds are depicted. PF stands for parameter-free.

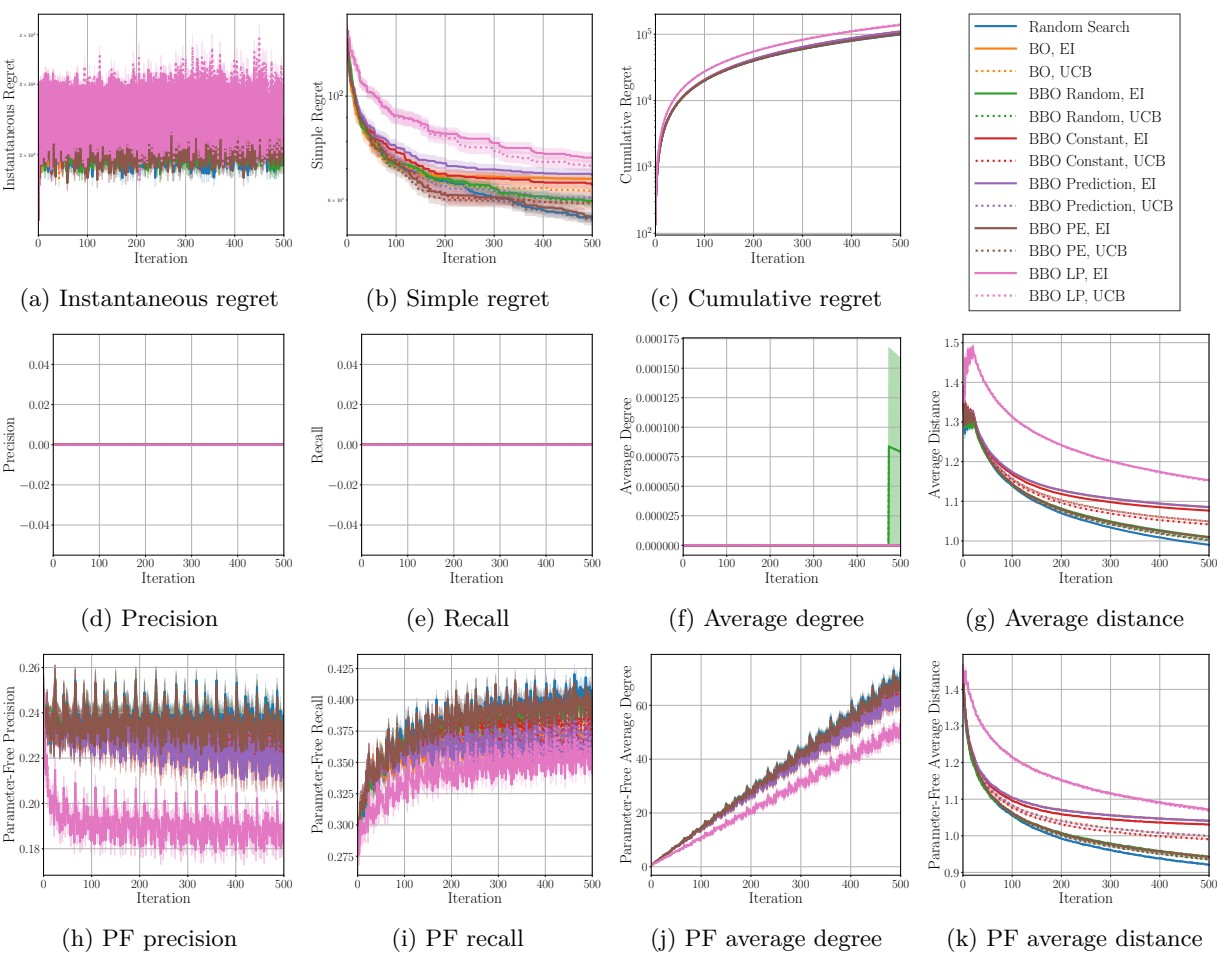

(a) Instantaneous regret

(b) Simple regret

(c) Cumulative regret

(d) Precision

(e) Recall

(f) Average degree

(g) Average distance

(h) PF precision

(i) PF recall

(j) PF average degree

(k) PF average distance

Figure 22: Bayesian optimization results versus iterations for the Levy 16D function. Sample means over 50 rounds and the standard errors of the sample mean over 50 rounds are depicted. PF stands for parameter-free.

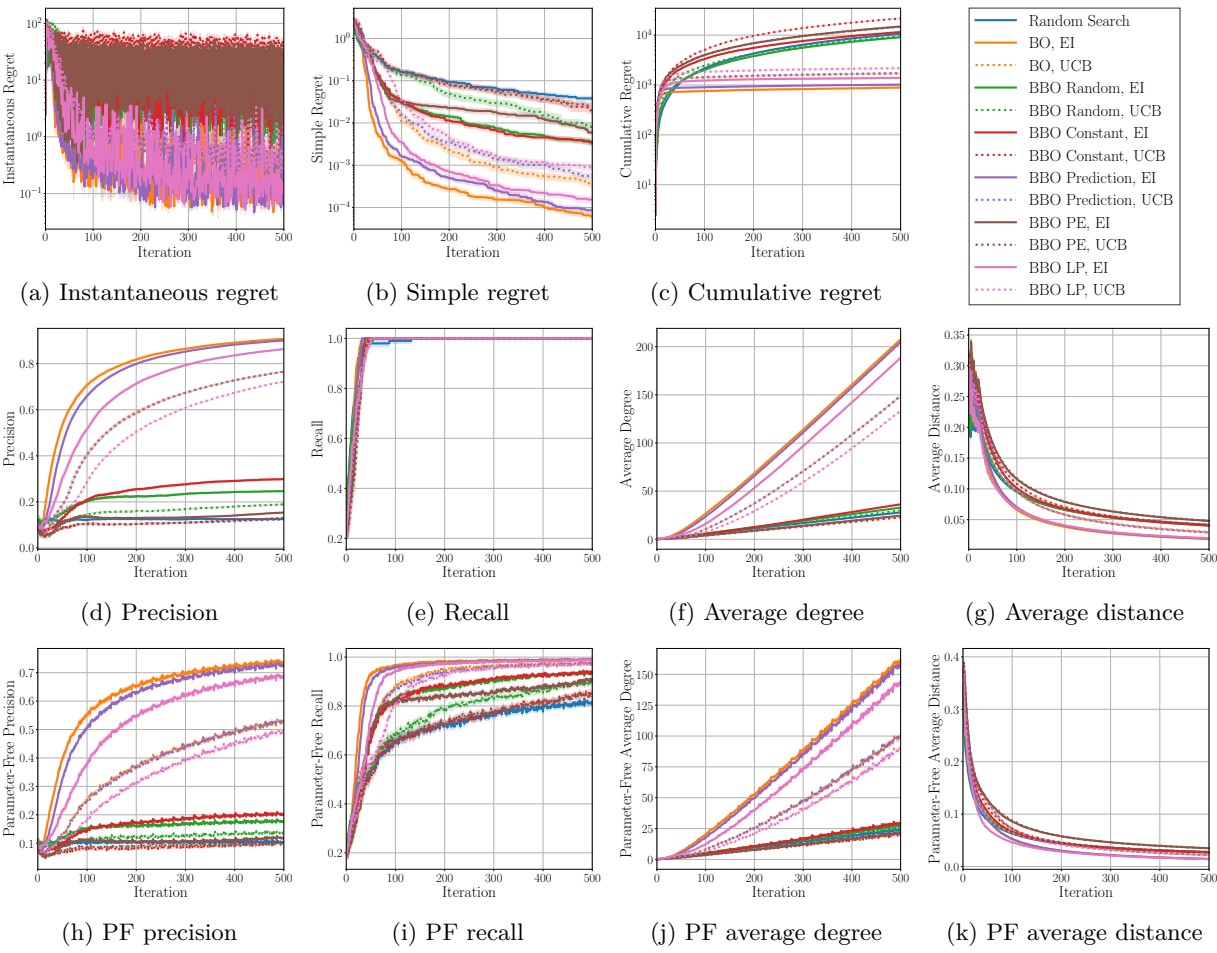

(a) Instantaneous regret     (b) Simple regret     (c) Cumulative regret

(d) Precision     (e) Recall     (f) Average degree     (g) Average distance

(h) PF precision     (i) PF recall     (j) PF average degree     (k) PF average distance

Figure 23: Bayesian optimization results versus iterations for the Six-Hump Camel function. Sample means over 50 rounds and the standard errors of the sample mean over 50 rounds are depicted. PF stands for parameter-free.

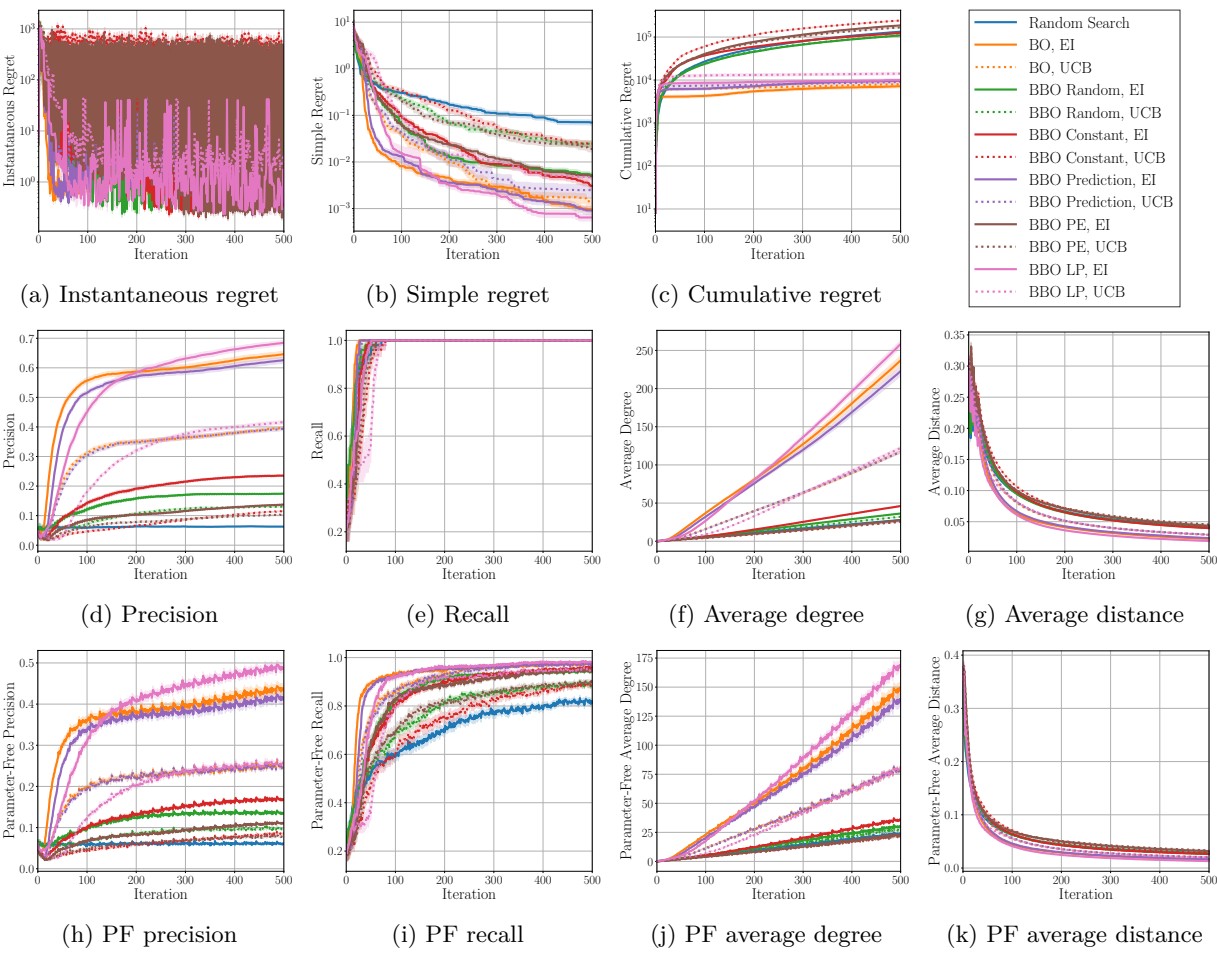

Figure 24: Bayesian optimization results versus iterations for the Three-Hump Camel function. Sample means over 50 rounds and the standard errors of the sample mean over 50 rounds are depicted. PF stands for parameter-free.

# C  Analysis on Metric Values in the Ackley 128D function

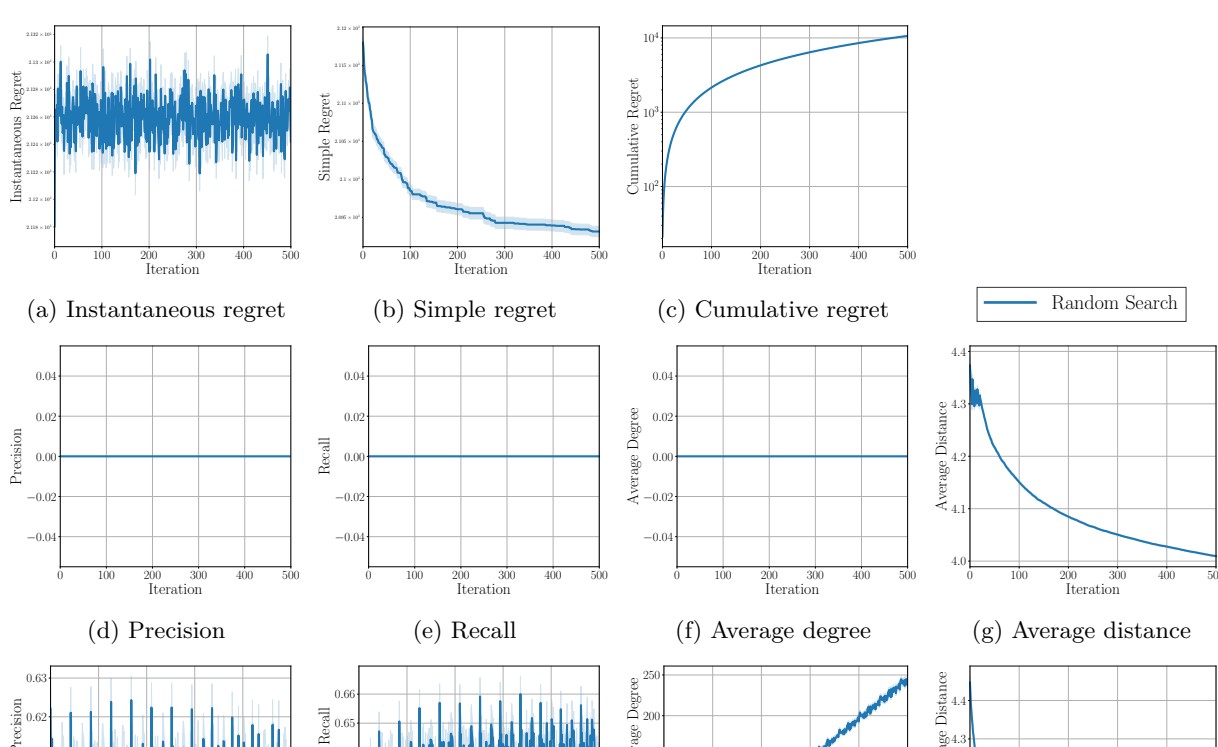

Figure 25: Optimization results versus iterations for the Ackley 128D function. Sample means over 50 rounds and the standard errors of the sample mean over 50 rounds are depicted. PF stands for parameter-free.

To show the behavior of the regret-based and geometric metrics in higher-dimensional problems, we test random search on the Ackley 128D function, which is considered as a high-dimensional problem in the Bayesian optimization community. According to this analysis presented in Figure 25, our parameter-free geometric metrics successfully represent the performance of an optimization algorithm in terms of distinct criteria, so that we can interpret the optimization results from geometric perspectives.

# D    Additional Analysis on the Spearman's Rank Correlation Coefficients between Metrics

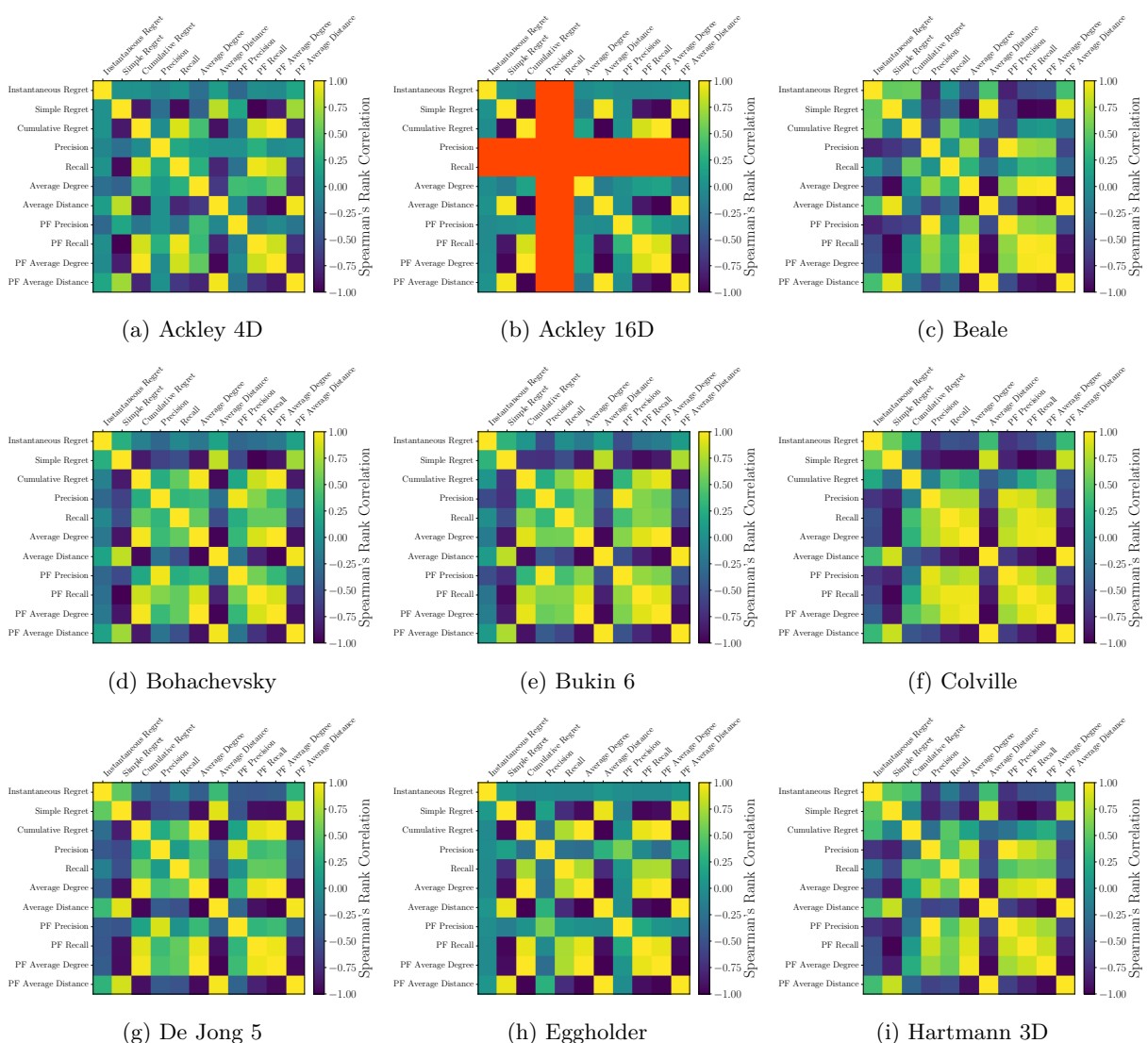

Figure 26: Spearman's rank correlation coefficients between metrics for different benchmark functions such as the Ackley 4D, Ackley 16D, Beale, Bohachevsky, Bukin 6, Colville, De Jong 5, Eggholder, and Hatmann 3D functions. Red regions indicate the coefficients with NaN values.

We visualize the Spearman's rank correlation coefficients between metrics for diverse benchmark functions in Figures 26 and 27. These results are obtained by using the same settings detailed in the main article.

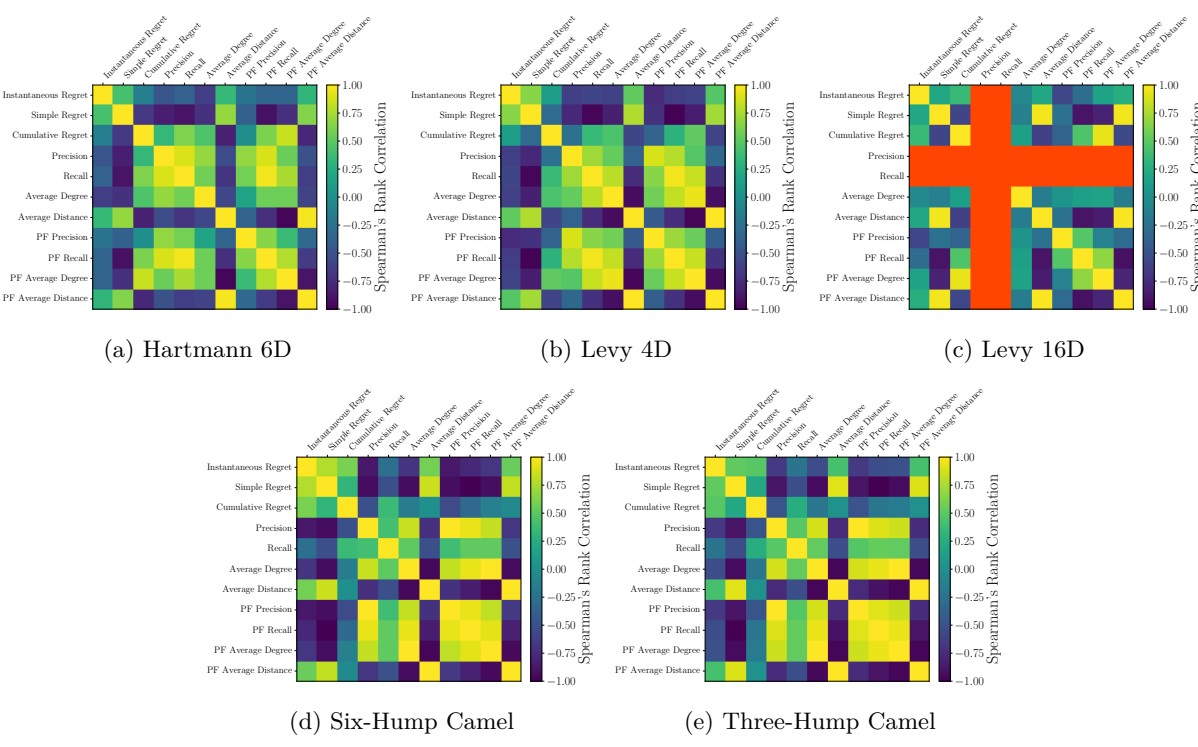

Figure 27: Spearman's rank correlation coefficients between metrics for different benchmark functions such as the Hartmann 6D, Levy 4D, Levy 16D, Six-Hump Camel, and Three-Hump Camel functions. Red regions indicate the coefficients with NaN values.

