# OpenReview forum: "Beyond Regrets: Geometric Metrics for Bayesian Optimization"
_TMLR — Rejected by TMLR_

### Review · Reviewer_Rhch · 2024-01-15

**Summary Of Contributions:**

In this paper, in addition to existing regret-based metrics, the authors look at complementary metrics to analyze Bayesian optimization (BO), via distance-based criteria between evaluated designs and global optima. A parameter free version is proposed based on a distribution. Several BO methods are run on various benchmark functions to empirically evaluate the correlation between the geometric- and regret-based metrics.

**Audience:**

Yes

**Claims And Evidence:**

No

**Requested Changes:**

A more precise justification for the definition of the new metrics is needed, in the context of global optimization. Just showing that there are correlations is insufficient.

A typical way of attributing iterations to exploration or exploitation is to look at the leading term in the expected improvement. This analysis could added as well.


Further clarifications

- Figure 1 and 2: the difference between the top and bottom is not clear, aren’t they showing function values rather than precision and recall?

- Section 4: Why not using [0, 1]^d? This would be easier to understand distances.

- A related paper is Wessing, S., & Preuss, M. (2017, November). The true destination of EGO is multi-local optimization. In 2017 IEEE Latin American Conference on Computational Intelligence (LA-CCI) (pp. 1-6). IEEE.

**Strengths And Weaknesses:**

Strengths
- The proposed criteria are easy to compute.
- Parameter-free versions are showcased. But they rely on an arbitrary distribution (hence they are not distribution free).

Weaknesses
- It is somehow an ill-posed problem: the initial goal is global optimization, while the goal of finding diverse solutions is not precisely described.
- Hence the justification for introducing new metrics is not sufficiently clear, and they do not take into account function values. Also, how realistic is it to suppose that there are several global optima?
- The metrics are not robust with respect to an increase in the number of variables.

---

> ### Author Response · Authors · 2024-01-24
>
> We thank the reviewer for the valuable comment.
>
> > It is somehow an ill-posed problem: the initial goal is global optimization, while the goal of finding diverse solutions is not precisely described.
>
> > Hence the justification for introducing new metrics is not sufficiently clear, and they do not take into account function values.
>
> It is not an ill-posed problem since our goal is not to completely replace regret-based metrics with the geometric metrics. With our metrics as well as the regret-based metrics, we can understand the results of Bayesian optimization more thoroughly. To justify needs for the geometric metrics, we have added more references in ***Section 1*** and explicitly specified the goal of our metrics in ***Sections 1, 2, and 3***. In particular, it is important to find more diverse solutions in real-world problems such as virtual screening and materials discovery. We would like to note that *Reviewer JZxR* has agreed with this point.
>
> > Also, how realistic is it to suppose that there are several global optima?
>
> Indeed, it is a good point.  However, regret-based metrics also share the similar drawback because they need to know the evaluation of global optima. We have added more thorough discussion on this topic in ***Section 5***. We think that approximating global solutions to calculate metrics can be an interesting future research topic. We would like to emphasize that our average degree, average distance, parameter-free average degree, and parameter-free average distance do not require such information, unlike the regret-based metrics and the other geometric metrics.
>
> > The metrics are not robust with respect to an increase in the number of variables.
>
> We think that this comment is provided based on the misunderstanding of our manuscript.  As shown in ***Figure 5***, our metrics, in particular, the parameter-free forms, are robust with respect to an increase in the number of variables.  More precisely, we have made our metrics robust to the number of variables by proposing the parameter-free metrics.
>
> > A more precise justification for the definition of the new metrics is needed, in the context of global optimization. Just showing that there are correlations is insufficient.
>
> By considering your comment, we have revised ***Section 3***. We added more detailed justifications on our geometric metrics and the sampling distributions for $M$.
>
> > A typical way of attributing iterations to exploration or exploitation is to look at the leading term in the expected improvement. This analysis could added as well.
>
> We have discussed your comment in ***Section 5***. Please see the corresponding section.
>
> > Figure 1 and 2: the difference between the top and bottom is not clear, aren’t they showing function values rather than precision and recall?
>
> We have updated their captions. They are showing function values and they illustrate how precision and recall are computed. In addition, we have moved Figure 2 to the appendices; Figure 2 is now Figure 10.
>
> > Section 4: Why not using $[0, 1]^d$? This would be easier to understand distances.
>
> Thank you for pointing this out.  We have normalized all search spaces to $[0, 1]^d$ and updated our manuscript accordingly.  As you said, understanding the underlying meaning of distances becomes easier; please see ***Section 4*** of the revision.
>
> > A related paper is Wessing, S., & Preuss, M. (2017, November). The true destination of EGO is multi-local optimization. In 2017 IEEE Latin American Conference on Computational Intelligence (LA-CCI) (pp. 1-6). IEEE.
>
> We have added the reference you recommended. Please see ***Section 5***.

---

> > ### Comment · Reviewer_Rhch · 2024-01-30
> >
> > Thank for providing additional elements and improving the paper.
> >
> > > It is not an ill-posed problem since our goal is not to completely replace regret-based metrics with the geometric metrics. With our metrics as well as the regret-based metrics, we can understand the results of Bayesian optimization more thoroughly. To justify needs for the geometric metrics, we have added more references in Section 1 and explicitly specified the goal of our metrics in Sections 1, 2, and 3. In particular, it is important to find more diverse solutions in real-world problems such as virtual screening and materials discovery. We would like to note that Reviewer JZxR has agreed with this point.
> >
> > My point is that diversity is not included in the global optimization problem. I agree that in practice someone would be interesting in having a choice of 'good' solutions from various parts of the input domain, but it requires properly defining what good and distant mean. It could be for instance based on local optima, or on some additional constraints. Still, this remains a problem formulation issue.
> >
> > Overall, it remains unclear to me that the geometry measures provide additional useful information. Take precision, I don't get why it should increase: once the GP model is good in one region, it should not be sampled anymore (e.g., for stability reasons or when considering non myopic infill criteria, see e.g., Osborne, M., & Osborne, M. A. (2010). Bayesian Gaussian processes for sequential prediction, optimisation and quadrature (Doctoral dissertation, Oxford University, UK).). Then average degree and average distance won't behave well in high-dimension.

---

> ### Author Response · Authors · 2024-02-01
>
> Thank you for your response.  We are pleased to discuss your constructive comment with the reviewer.
>
> > My point is that diversity is not included in the global optimization problem.
>
> We think that we can not agree with this comment.
>
> First off, in general, we mention that Bayesian optimization has an ability to explore and exploit a search space. This ability is directly related to diversity. However, we do not measure the degrees of exploration and exploitation and only care about the regret-based metrics. Since we do not measure them, it is hard to ensure that Bayesian optimization algorithms are exploring well or exploiting well. To demonstrate this ability, we necessitate defining new metrics to measure the degrees of exploration and exploitation. As a result, our geometric metrics can be employed to numerically represent the trade-off of exploration and exploitation through Bayesian optimization.
>
> Furthermore, Bayesian optimization is generally used to solve global optimization problems, but we do not need to confine Bayesian optimization to particular problems. We can include the consideration of diversity in the global optimization problems. For example, we can solve a problem to find global solution candidates and simultaneously diversify the candidates found by Bayesian optimization -- as mentioned in our previous response and the revision, such a problem is motivated by many scientific and engineering problems.
>
> > I agree that in practice someone would be interesting in having a choice of 'good' solutions from various parts of the input domain, but it requires properly defining what good and distant mean.
>
> Thank you for agreeing the need to find diversified solutions. To measure how good and distant the solutions found are, we defined geometric metrics and empirically showed the validity of our metrics.
>
> > It could be for instance based on local optima, or on some additional constraints. Still, this remains a problem formulation issue.
>
> We believe that our metrics can be used in the problems you mentioned, in order to verify if the problems are successfully solved.
>
> > Overall, it remains unclear to me that the geometry measures provide additional useful information. Take precision, I don't get why it should increase: once the GP model is good in one region, it should not be sampled anymore.
>
> We think that your comment is not true in Bayesian optimization.
>
> Firstly, defining a good GP model is almost infeasible in practical scenarios. If we can access a good GP model, we do not need to run Bayesian optimization. We can choose the best point from the GP model instead.
>
> Secondly, the general goal of Bayesian optimization is to balance exploration and exploitation. The definition of exploitation means that we frequently evaluate the region closer to the points evaluated. Our metrics are capable of measuring this property numerically.
>
> Thirdly, observation noises can exist in practical examples. Sometimes, we need to repeatedly evaluate the points close to the points we have already evaluated, in order to identify the observation noises.
>
> > (e.g., for stability reasons or when considering non myopic infill criteria, see e.g., Osborne, M., & Osborne, M. A. (2010). Bayesian Gaussian processes for sequential prediction, optimisation and quadrature (Doctoral dissertation, Oxford University, UK).)
>
> Could you explicitly point out where they are mentioned? We could not find the reasons you mentioned. You simply referred to the Michael Osborne's dissertation of over 200 pages.
>
> > Then average degree and average distance won't behave well in high-dimension.
>
> We have already shown the results on three 16-dimensional examples; see ***Figures 4, 12, and 22***. They behaved well. Moreover, by considering your comment, to show the behavior of our metrics in a higher-dimensional case, we have added new experiments on the 128-dimensional problem to optimize the Ackley 128D function; please see ***Appendix C*** and ***Figure 25***. Our parameter-free metrics still work appropriately.

---

> > ### Comment · Reviewer_Rhch · 2024-02-02
> >
> > > First off, in general, we mention that Bayesian optimization has an ability to explore and exploit a search space. This ability is directly related to diversity. However, we do not measure the degrees of exploration and exploitation and only care about the regret-based metrics. Since we do not measure them, it is hard to ensure that Bayesian optimization algorithms are exploring well or exploiting well. To demonstrate this ability, we necessitate defining new metrics to measure the degrees of exploration and exploitation. As a result, our geometric metrics can be employed to numerically represent the trade-off of exploration and exploitation through Bayesian optimization.
> >
> > > Furthermore, Bayesian optimization is generally used to solve global optimization problems, but we do not need to confine Bayesian optimization to particular problems. We can include the consideration of diversity in the global optimization problems. For example, we can solve a problem to find global solution candidates and simultaneously diversify the candidates found by Bayesian optimization -- as mentioned in our previous response and the revision, such a problem is motivated by many scientific and engineering problems.
> >
> > Different acquisition functions target different goals, with a corresponding exploration-exploitation tradeoff. Some may look at level sets, see e.g., Bogunovic, I., Scarlett, J., Krause, A., & Cevher, V. (2016). Truncated variance reduction: A unified approach to bayesian optimization and level-set estimation. Advances in neural information processing systems, 29. or global accuracy, e.g., Leatherman, E. R., Santner, T. J., & Dean, A. M. (2018). Computer experiment designs for accurate prediction. Statistics and Computing, 28, 739-751.
> >
> > For global optimization, the focus is on finding *a* design with minimum value. Finding diverse local minima is a different problem.
> > I do not see a direct link with optimizing precision/recall/average degree or distance. Especially since in practice we do not have access to global optima locations to compute some of them.
> >
> > > We believe that our metrics can be used in the problems you mentioned, in order to verify if the problems are successfully solved.
> > What do you mean by solved?
> >
> > > Secondly, the general goal of Bayesian optimization is to balance exploration and exploitation. The definition of exploitation means that we frequently evaluate the region closer to the points evaluated. Our metrics are capable of measuring this property numerically.
> >
> > The goal of BO is to find $\mathbf{x} \in \arg \min(f(\mathbf{x})$). What do I gain from measuring exploitation? I can exploit close to a bad local optima, or over explore, this is problem dependent.
> >
> > > Could you explicitly point out where they are mentioned? We could not find the reasons you mentioned. You simply referred to the Michael Osborne's dissertation of over 200 pages.
> >
> > This can be found in particular in Chapter 6.
> >
> > > We have already shown the results on three 16-dimensional examples; see Figures 4, 12, and 22. They behaved well. Moreover, by considering your comment, to show the behavior of our metrics in a higher-dimensional case, we have added new experiments on the 128-dimensional problem to optimize the Ackley 128D function; please see Appendix C and Figure 25. Our parameter-free metrics still work appropriately.
> >
> > Indeed you obtain curves, with correlations but this is not causation and it is not clear to me that this is the behavior we would like to get.

---

> ### Author Response · Authors · 2024-02-02
>
> Thank you for your comment.
>
> We think that the reviewer is also mentioning the need for new metrics instead of regret-based metrics. We agree with this point.  Please let us quote your comment:
>
> > Different acquisition functions target different goals
>
> > I can exploit close to a bad local optima, or over explore, this is problem dependent.
>
> Some researchers may want to propose different acquisition functions to address different goals, or want to solve exploration-focused or exploitation-focused problems. But, they did not compute how different their algorithms numerically are. In response, our metrics can serve as measures for comparing the different goals of different acquisition functions and calculating the degrees of exploration and exploitation.
>
> > Finding diverse local minima is a different problem.
>
> In this paper, we are interested in problems with multiple global solutions.  It is not about finding diverse local minima.
>
> > I do not see a direct link with optimizing precision/recall/average degree or distance.
>
> We do not optimize them. They are used for measuring the performance of Bayesian optimization algorithms.
>
> > Especially since in practice we do not have access to global optima locations to compute some of them.
>
> We have already answered this comment in the previous response.  Regret-based metrics also share the same issue.

---

### Review · Reviewer_PirQ · 2024-01-15

**Summary Of Contributions:**

This paper proposes new metrics for evaluating the performance of Bayesian optimization, which are alternatives to the traditional regret-based metrics. The new metrics are able to measure the ability of a BO algorithm to find multiple global optima and characterize the exploration-exploitation behavior of a BO algorithm. The paper additionally proposes parameter-free variants of the metrics which integrate out the free parameters in the original metrics.

**Audience:**

Yes

**Broader Impact Concerns:**

The potential broader impact has been discussed.

**Claims And Evidence:**

Yes

**Requested Changes:**

- In equations 5 and 6, I guess the symbol $\vee$ represents the max operation? I think this should be clearly defined.
- Regarding average distance (equation 8), I think a natural idea is to set $k=t$, which helps get rid of the free parameter of $k$. Will this be a reasonable choice?
- Please also refer to the weaknesses I discussed above to see the corresponding requested changes.

**Strengths And Weaknesses:**

Strengths:
- The idea of the paper is very interesting, especially the importance of the ability of a metric to measure whether a BO algorithm can find multiple global optima.

Weaknesses:
- An important concern I have is how do we measure these new metrics in practice? In the experiments from the paper, they can be measured because the function to be optimized are synthetic functions for which we know the location of the global optima. However, in practice, we don't have information regarding the number and locations of the global optima, therefore, how should we calculate these metrics when we optimize a black-box function? I understand that regret also cannot be calculated in practice when we don't know the value of the global optimum, however, the definitions of regrets provide us useful means to perform theoretical analysis of an algorithm. Therefore, in order to make the proposed metrics more useful, some rigorous theoretical analysis should be performed, which the paper unfortunately has not included.
- I'm not sure how much practical value the average degree and average distance metrics actually provide. If I understand correctly, the paper has presented these metrics for evaluating the performance of a BO algorithm. I think it may make these two metrics more useful if they are repurposed for other use cases, e.g., maybe they can be used to decide when we should early-stop a BO algorithm (when the queried points start to concentrate in a small local region, it's probably time to stop the BO algorithm), or to provide interpretability for the decisions made by BO (i.e., every input selected by BO).
- I think some of the illustrations in the paper may not provide enough valuable insights to warrant being included in the main paper. For example, I think the interpretations offered by Fig. 1 and Fig. 2 can in fact be inferred from the metric definitions and hence it may not be necessary to use figures to illustrate them.
- Although the parameter-free versions of the metrics are supposed to help get rid of the dependency on free parameters, however, I think they may in fact introduce new parameters. For example, when sampling $\delta$ using the exponential distribution, how should we choose the rate parameter $1\d$? When sampling $k$ using a geometric distribution, why is the success rate parameter set at $0.5$? Perhaps some ablation study should be done to evaluate the impact of different choices of these hyperparameters.
- I think the writing of the paper can also be improved, because some places are not explained with perfect clarity. For example, at the bottom of page 9 when describing baseline (iii), I suppose the procedure described here is repeated for every batch (instead of the entire algorithm)?
- It has been claimed that in Figure 5, the parameter-free versions of precision and recall successfully compare the different algorithms. However, it's unclear to me how we can tell that the comparisons are successful. I think some more explanations are needed. Overall, I think it would make the paper more valuable if more insights/discussions are given to all empirical results in the paper.

---

> ### Author Response · Authors · 2024-01-24
> **Official Comment by Authors (1/2)**
>
> We thank the reviewer for the valuable comment.
>
> > An important concern I have is how do we measure these new metrics in practice? In the experiments from the paper, they can be measured because the function to be optimized are synthetic functions for which we know the location of the global optima. However, in practice, we don't have information regarding the number and locations of the global optima, therefore, how should we calculate these metrics when we optimize a black-box function? I understand that regret also cannot be calculated in practice when we don't know the value of the global optimum, however, the definitions of regrets provide us useful means to perform theoretical analysis of an algorithm. Therefore, in order to make the proposed metrics more useful, some rigorous theoretical analysis should be performed, which the paper unfortunately has not included.
>
> Indeed, it is a good point.  As you mentioned, regret-based metrics also share the similar drawback. They are used in the theoretical analysis of the convergence of Bayesian optimization though. We have added more thorough discussion on this topic in ***Section 5***. We think that approximating global solutions to calculate metrics can be an interesting future research topic. We would like to emphasize that our average degree, average distance, parameter-free average degree, and parameter-free average distance do not require such information, unlike the regret-based metrics and the other geometric metrics.
>
> > I'm not sure how much practical value the average degree and average distance metrics actually provide. If I understand correctly, the paper has presented these metrics for evaluating the performance of a BO algorithm. I think it may make these two metrics more useful if they are repurposed for other use cases, e.g., maybe they can be used to decide when we should early-stop a BO algorithm (when the queried points start to concentrate in a small local region, it's probably time to stop the BO algorithm), or to provide interpretability for the decisions made by BO (i.e., every input selected by BO).
>
> We think that your idea on the use of average degree and average distance is very compelling. We believe that automatic termination for Bayesian optimization or hyperparameter optimization has attracted much attention of researchers in this field. We have updated ***Sections 3.3 and 3.4*** in order to detail them. Because we think that the discussion on their potential use is out of the scope of this work, we do not include it in the current manuscript.
>
> > I think some of the illustrations in the paper may not provide enough valuable insights to warrant being included in the main paper. For example, I think the interpretations offered by Fig. 1 and Fig. 2 can in fact be inferred from the metric definitions and hence it may not be necessary to use figures to illustrate them.
>
> We have updated their captions. Also, we have moved Figure 2 to the appendices; Figure 2 is now Figure 10.
>
> > Although the parameter-free versions of the metrics are supposed to help get rid of the dependency on free parameters, however, I think they may in fact introduce new parameters. For example, when sampling $\delta$ using the exponential distribution, how should we choose the rate parameter $1 / d$? When sampling $k$ using a geometric distribution, why is the success rate parameter set at 0.5? Perhaps some ablation study should be done to evaluate the impact of different choices of these hyperparameters.
>
> Thank you for pointing this out. We have added a more detailed justification on why these distributions are selected in ***Section 3.5***. To sum up, we selected them because of the monotonically decreasing property of both distributions. Please see ***Section 3.5*** for details. Moreover, in our preliminary experiments, the parameters for the exponential and geometric distributions are not sensitive; they changed metric values but did not affect the tendency of metric values.  On the other hand, the dimensionality of an optimization problem is only a significant factor. Thus, we consider $d$ for the exponential distribution.
>
> Furthermore, the most sensitive parameter was the number of samples $M$. In response, we conducted new experiments on showing the impact of $M$. As presented in ***Figure 9***, if $M$ is equal to or greater than 100, the oscillation of metric values is not a serious issue. We have added ***Section 5*** to discuss this topic.
>
> > I think the writing of the paper can also be improved, because some places are not explained with perfect clarity.
>
> Thank you for pointing this out. We have revised our manuscript in order to improve our writing.
>
> > For example, at the bottom of page 9 when describing baseline (iii), I suppose the procedure described here is repeated for every batch (instead of the entire algorithm)?
>
> Yes, that procedure is repeated for every batch. We have updated our paper accordingly.

---

> ### Author Response · Authors · 2024-01-24
> **Official Comment by Authors (2/2)**
>
> > It has been claimed that in Figure 5, the parameter-free versions of precision and recall successfully compare the different algorithms. However, it's unclear to me how we can tell that the comparisons are successful. I think some more explanations are needed. Overall, I think it would make the paper more valuable if more insights/discussions are given to all empirical results in the paper.
>
> By considering your comment, we have revised ***Sections 3 and 4*** and added ***Section 5***. We tried to provide more insights and discussions; please let us know if our manuscript is still unclear.
>
> > In equations 5 and 6, I guess the symbol $\vee$ represents the max operation? I think this should be clearly defined.
>
> It is a logical disjunction (or logical or) operation. We have described it in the updated manuscript.
>
> > Regarding average distance (equation 8), I think a natural idea is to set $k = t$, which helps get rid of the free parameter of $k$. Will this be a reasonable choice?
>
> We think that $k$ should be sufficiently smaller than $t$.
>
> Suppose that we are given 4 query points. We will assume two scenarios where average distance (without considering the nearest neighbors) from one point to the other points is $d$. Note that when we obtain $k$ nearest neighbors, we assume that it includes a reference point itself. For example, if $k = 4$, the nearest neighbors will be all four points.
>
> The first scenario is that two points are identical and the other points, which are also identical, are far from two points. If $k = t = 4$, the average distance will be still $d$. But, if $k = 2$, the average distance will be $0$.
>
> The second scenario is that each of four points is far from the other points. If $k = t = 4$, the average distance will be $d$. On the other hand, if $k = 2$, the average distance will be between $0$ and $d$.
>
> From the perspective of measuring the degree of exploration, the examples with $k = 2$ is more reasonable. Therefore, we should set $k$ as a value smaller than $t$.

---

### Review · Reviewer_JZxR · 2024-01-22

**Summary Of Contributions:**

This work is concerned with the problem of studying optimization performance of Bayesian optimization algorithms. Four new measures (precision, recall, average degree, and average distance) are introduced to quantitatively compute the optimization performance. All these measures capture a metric over observed input locations as opposed to the output objective values commonly used in simple/cumulative regret. Experiments are performed on multiple synthetic functions (with known global optima) to analyze these measures.

**Audience:**

Yes

**Claims And Evidence:**

No

**Requested Changes:**

See the review comments.

**Strengths And Weaknesses:**

- I like the fact that the paper talks about an interesting problem setting related to how modern BO is used. Many modern BO algorithms are aimed toward finding multiple global/local optima. This is relevant for many real-world settings (like drug discovery) since we would like the BO procedure to give us many different potential solutions to hedge our bets on downstream tasks.

- One key question I have is about the practical use of the proposed metrics. In most real-world applications, we never have the global optima available to us. Therefore, most BO papers just report the best observed value as a function of time to show their performance. This is an issue with existing metrics like simple regret/cumulative regret as well but both of them are introduced in papers showing theoretical convergence guarantees. Since this paper is only concerned with the definition/implementation of these metrics (and not any theoretical analysis), it is important to motivate how can we use them for real-world problems.

- Since each of the metrics require a parameter (ball radius \delta etc.) to define them, the paper provides another approach where we take an expectation over these parameters to define "parameter-free" metrics. This seems reasonable but it requires defining a distribution over these parameters now. The paper picks exponential distribution for precision, recall, average degree measures and geometric distribution for average distance measure without any justification. This is a critical choice and needs to be discussed appropriately.

- In the experiment section, the paper mentions that "One drawback of our parameter-free metrics excluding the parameter-free average distance is that their values are prone to oscillating.". This drawback is captured by the no. of monte carlo samples (M) for computing the expectation in "parameter-free" version of the proposed metrics. This seems slightly less appealing since we are replacing one parameter with another sensitive parameter (please note that this oscillation happens even at a large value of M=1000).

---

> ### Author Response · Authors · 2024-01-24
>
> We thank the reviewer for the valuable comment.
>
> > I like the fact that the paper talks about an interesting problem setting related to how modern BO is used. Many modern BO algorithms are aimed toward finding multiple global/local optima. This is relevant for many real-world settings (like drug discovery) since we would like the BO procedure to give us many different potential solutions to hedge our bets on downstream tasks.
>
> We are happy that you like our paper.  We have added more references in ***Section 1*** by considering your comment. Please see the first paragraph of ***Section 1***.
>
> > One key question I have is about the practical use of the proposed metrics. In most real-world applications, we never have the global optima available to us. Therefore, most BO papers just report the best observed value as a function of time to show their performance. This is an issue with existing metrics like simple regret/cumulative regret as well but both of them are introduced in papers showing theoretical convergence guarantees. Since this paper is only concerned with the definition/implementation of these metrics (and not any theoretical analysis), it is important to motivate how can we use them for real-world problems.
>
> Indeed, it is a good point.  As you mentioned, regret-based metrics also share the similar drawback. They are used in the theoretical analysis of the convergence of Bayesian optimization though. We have added more thorough discussion on this topic in ***Section 5***. We think that approximating global solutions to calculate metrics can be an interesting future research topic. We would like to emphasize that our average degree, average distance, parameter-free average degree, and parameter-free average distance do not require such information, unlike the regret-based metrics and the other geometric metrics.
>
> > Since each of the metrics require a parameter (ball radius $\delta$ etc.) to define them, the paper provides another approach where we take an expectation over these parameters to define "parameter-free" metrics. This seems reasonable but it requires defining a distribution over these parameters now. The paper picks exponential distribution for precision, recall, average degree measures and geometric distribution for average distance measure without any justification. This is a critical choice and needs to be discussed appropriately.
>
> Thank you for pointing this out. We have added a more detailed justification on why these distributions are selected in ***Section 3.5***. To sum up, we selected them because of the monotonically decreasing property of both distributions. Please see ***Section 3.5*** for details.
>
> > In the experiment section, the paper mentions that "One drawback of our parameter-free metrics excluding the parameter-free average distance is that their values are prone to oscillating.". This drawback is captured by the no. of monte carlo samples ($M$) for computing the expectation in "parameter-free" version of the proposed metrics. This seems slightly less appealing since we are replacing one parameter with another sensitive parameter (please note that this oscillation happens even at a large value of $M=1000$).
>
> First off, we made a typo. We used $M = 100$. Nevertheless, $M$ becomes less sensitive to metric values if $M$ is sufficiently larger. To show its insensitiveness, we conducted new experiments on showing the impact of $M$. As presented in ***Figure 9***, if $M$ is equal to or greater than 100, the oscillation of metric values is not a serious issue. We have updated this paragraph based on the new experiments; the paragraph is moved to ***Section 5***.

---

### Author Response · Authors · 2024-01-24
**General Comment to All Reviewers**

We deeply appreciate the reviewers' constructive feedback to improve our work.

The reviewers provided the strengths of our work in their comments. In particular, *Reviewer JZxR* mentioned that ***I like the fact that the paper talks about an interesting problem setting related to how modern BO is used***, ***Many modern BO algorithms are aimed toward finding multiple global/local optima***, and ***This
is relevant for many real-world settings (like drug discovery***.
*Reviewer PirQ* commented that ***The idea of the paper is very interesting, especially the importance of the ability of a metric to measure whether a BO algorithm can find multiple global optima***.
*Reviewer Rhch* mentioned that ***The proposed criteria are easy to compute*** and ***Parameter-free versions are showcased***.

Based on the reviewers' suggestions, we have made the following improvements to our paper:

* Revised ***Abstract***
* Added more references in ***Section 1*** to justify our metrics
* Specified the goal of our geometric metrics in ***Sections 1, 2, and 3***
* Moved ***Figure 2*** to the appendices; it is now ***Figure 10***
* Added more detailed justifications of our metrics in ***Section 3***.
* Normalized all search spaces to $[0, 1]^d$ and recalculated metric values and correlations
* Added a new figure ***Figure 9*** to show the impact of the number of samples
* Discussed related work, parameter-free metrics, global solution information in ***Section 5***
* Improved the writing of our manuscript

We will answer the individual comments in the respective replies.

---

> ### Author Response · Authors · 2024-01-29
>
> Dear reviewers,
>
> We tried to answer your questions and concerns in the previous responses.
>
> Please let us know if you have any further questions.
>
> Sincerely,
>
> Author(s) of the submission 2010.

---

### Decision · Action_Editor_Fc56 · 2024-03-05

**Recommendation:** Reject

**Comment:**

This manuscript concerns Bayesian optimization, a popular framework for the optimization of expensive-to-evaluate objective functions. The focus of the manuscript is the proposal for several new metrics for evaluating Bayesian optimization routines that attempt to give more consideration to the exploration of multiple global optima, something that typical metrics based on regret are not equipped to do.

Although the reviewers agree that this material is of interest to TMLR's audience, they also identified several perceived weaknesses with the manuscript as submitted. namely:

- some proposed metrics may be challenging to use in practice as they require knowledge that is typically unavailable in order to evaluate
- some proposed metrics specifically target the case of having multiple global optima, which may not be a particularly notable concern in practice
- uncertainty regarding the justification of certain technical choices made by the authors in their algorithmic design

Ultimately, despite significant engagement during the author-reviewer discussion period, these issues were left unresolved and deemed too serious to justify publication of the submitted manuscript. As these issues bring into question the very foundation of the submission, it is not clear whether even major revisions would suffice to adequately address them.

**Audience:**

Yes, the manuscript focuses on the popular framework of Bayesian optimization, which is of interest to a nontrivial subset of TMLR's audience.

**Claims And Evidence:**

No, the reviewers did not reach consensus that the claims made in this submission are supported by clear and convincing evidence. In particular, the reviewers identified several perceived weaknesses with the submitted manuscript that could not be adequately resolved during the author-reviewer discussion period. I elaborate on these issues below.